# Probabilistic DiffusionNet: A geometry informed probabilistic generative surrogate for PDEs

## Abstract

We propose a probabilistic generative extension of the DiffusionNet architecture, widely used for surface learning tasks, by introducing latent random variables derived from a stochastic reformulation of the underlying diffusion process. The resulting probabilistic model can be used as a resolution-invariant and uncertainty-aware surrogate for the trace solution map of PDEs whose boundary conditions are determined by surface geometry. Such a surrogate can expedite and inform typical engineering design and optimisation processes that require computationally burden-some computational fluid dynamics (CFD) analysis pipelines. We demonstrate that the proposed architecture produces superior uncertainty quantification (UQ) performance on standard CFD datasets without sacrificing predictive accuracy, while enjoying lower computational cost compared to other prevalent geometry-informed PDE surrogates endowed with UQ capabilities.

## 1 Introduction

Across science and engineering it is often needed to produce solutions to non-linear partial differential equations (PDEs) with boundary conditions, as they describe the behaviour of a wide range of physical phenomena, from fluid dynamics and hydro-electromagnetics to relativistic gravitation. During the typical design process it is necessary to solve these PDEs for changing geometric conditions, which puts a time-consuming numerical solver in a loop. Machine learning promises to expedite the optimisation lifecycle by introducing models as surrogates for PDE solvers (Li et al., 2023d; Wu et al., 2024; Oldenburg et al., 2022; Sharp et al., 2022). For reliable application, a surrogate should to be: (i) *invariant to the resolution* of the chosen discretisation, and (ii) able to produce *predictive uncertainty*. The latter is particularly important in engineering applications where the output of a surrogate is used to inform downstream decision-making.

Operator learning is a popular paradigm for building surrogate models that learn mappings between function spaces (Lu et al., 2019; Li et al., 2020a; Kovachki et al., 2023). Neural Operators, a type of operator learning, approximate a PDE's solution operator, enabling generalization across resolutions. However, methods like the Fourier Neural Operator (FNO) (Li et al., 2021) and Geometry-Informed Neural Operator (GINO) (Li et al., 2023d), when applied to problems with geometric boundary conditions, do not adequately leverage geometric information. This leads to overparameterised networks and inefficient uncertainty quantification, as they fail to incorporate useful geometric inductive biases.

In contrast, geometric deep learning architectures (Eliasof et al., 2021; Wu et al., 2024; Charles et al., 2017a; Bamberger et al., 2024) learn a mapping from 3D geometry to a PDE solution on the geometry's surface. These methods are parameter-efficient due to geometric inductive biases but can be overly sensitive to mesh discretization and struggle with global interactions. Recently, Sharp et al. (2022) introduced DiffusionNet (DN) to address both of these issues. DiffusionNet draws its strength for various surface learning tasks by using two simple geometric operations: (i) a diffusion layer with learnable diffusion time parameters for message-passing, and (ii) spatial gradient features according to manifold orientation for capturing anisotropy. Importantly, both these operations are agnostic to the choice of discretisation and sampling, and thus the resulting architecture is resolution invariant by construction, akin to a Neural Operator.

**Contributions.** In this work, we develop a probabilistic reformulation of DiffusionNet using a Fourier expansion of stochastic partial differential equations (SPDEs) (Holden et al., 1996; Walsh, 1986). This approach encodes uncertainty directly through the message-passing mechanism, avoiding the familiar limitations such as poor uncertainty quality, drop in accuracy, and lack of computational scalability of traditional Bayesian Neural Networks (Hernández-Lobato & Adams, 2015; Blundell et al., 2015) or post-hoc methods like Laplace approximation (MacKay, 1992; Ritter et al., 2018) and Conformal Predictions (Vovk et al., 2005). Instead, our reformulation allows us to build a probabilistic generative model, Probabilistic DiffusionNet (PDN), in the same spirit as a traditional variational auto-encoder that allows us to estimate PDE solutions given varying boundaries, with spatially correlated probabilistic outputs on those surfaces that reflect uncertainties. Our experiments show, firstly, that DiffusionNet is a powerful surrogate model competitive with state-of-the-art (SOTA) Neural Operators and PDE surrogates. We then show that PDN retains DiffusionNet's predictive performance while producing well-calibrated uncertainty estimates compared to standard post-hoc uncertainty quantification methods.

## 1.1 RELATED WORK

In this section we review the classes of models that have dominated the space of PDE surrogates, and approaches to endow them with uncertainty quantification. We provide an extended discussion on related works in Appendix A and F.

**Geometric Deep Learning** Graph Neural Graph Neural Networks (GNNs) (Scarselli et al., 2009; Sanchez-Gonzalez et al., 2020; Niepert et al., 2016; Li & Farimani, 2022) use message-passing to predict fields on mesh-structured data (Dalton et al., 2022; Pfaff et al., 2020; Horie & Mitsume, 2022; Brandstetter et al., 2022; Gladstone et al., 2024). Recently, Sharp et al. (2022) introduced DiffusionNet, which employs diffusion and gradient mechanisms for message-passing, building on the idea of GNNs as solutions to diffusion equations (Chamberlain et al., 2021).

For uncertainty quantification (UQ) in GNNs, methods include frequentist approaches like temperature-scaling (Guo et al., 2017; Wang et al., 2021), conformal predictions (Vovk et al., 2005; Huang et al., 2023a), ensembling (von Pichowski et al., 2024; Lin et al., 2022; Xu et al., 2022), as well as Bayesian methods (Lamb & Paige, 2020; Hasanzadeh et al., 2020). More recently, Lin et al. (2024b) and Bergna et al. (2024) extended GNNs by modeling stochastic diffusion/ODE processes to capture uncertainty. However, these methods require complex backpropagation through numerical solvers, making them computationally unsuitable for large-scale applications like CFD.

Neural Operators (NOs) (Kovachki et al., 2023; Lu et al., 2019; Pepe et al., 2025) are a popular method for learning function space maps. A key class of NOs, like FNO, capture long-range correlations using basis decompositions, with variants exploring alternative bases (Tripura & Chakraborty, 2023; Gupta et al., 2021; Bonev et al., 2023; Cao et al., 2024; Chen et al., 2024). GINO combines the graph-based GNO (Li et al., 2020a;b) for irregular domains with the FNO for regular domains. Other work has focused on alternative message-passing schemes (Li et al., 2023a; Alkin et al., 2024; Kissas et al., 2022).

For uncertainty quantification (UQ) in NOs, existing methods often require expensive post-training steps, such as when using the Laplace approximation (Magnani et al., 2022), generative models with Monte Carlo methods (Meng et al., 2022), or conformal prediction extensions (Ma et al., 2024b) Taking a different approach, Salvi et al. (2022) propose a Neural SPDE method similar to the ideas in Lin et al. (2024a) and Bergna et al. (2024), but lacks a direct UQ objective and requires slow numerical solvers. Our work combines the SPDE formalism with variational inference and the geometric inductive bias of DiffusionNet to create a scalable architecture for natural uncertainty quantification.

**Point Cloud Methods** Given the ubiquity of point-cloud representations in engineering and graphics applications, several methods (Charles et al., 2017a;b) have emerged that do not rely on connectivity or grid structures. Recent approaches (Guo et al., 2021; Xiao et al., 2024) adapt transformers to point-clouds, offering strong but computationally intensive PDE surrogates. To address the quadratic complexity of attention mechanisms, Cao (2021) propose the Galerkin Transformer, while Wu et al. (2024) introduce a Physics Attention layer. Both approaches scale linearly with the number of input

points. While native uncertainty quantification remains unresolved, classical techniques like MC dropout and Laplace approximation are applicable.

## 2 BACKGROUND

### 2.1 PROBLEM SETTING

Our work addresses problems, such as design optimisation, where multiple geometric shapes are evaluated by solving systems of partial differential equations (PDEs). Formally, we consider a space of shapes $\mathcal{S}$, where each shape corresponds to a surface $M_s$ embedded in $\mathbb{R}^3$. For each shape, we consider a PDE system:

$$\begin{cases} \mathcal{L}(u_s) = f & \text{in } \Omega_s \\ \mathcal{B}(u_s) = b & \text{on } \partial\Omega_s \end{cases} \tag{1}$$

where $u_s : \overline{\Omega}_s \to \mathbb{R}^{d_u}$ is the solution field and $\Omega_s$ is the domain surrounding the surface $M_s$. $\mathcal{L}$ and $\mathcal{B}$ are differential and boundary operators. We assume this PDE is well-posed, meaning a unique solution $u_s$ exists for any valid shape $s$ and boundary conditions $b$ and forcing function $f$. In many applications—aerodynamics, heat transfer, structural mechanics—the quantities of interest are surface fields obtained via the trace operator:

$$u_s^\dagger = T_s(u_s) : M_s \to \mathbb{R}^{d_u}$$

We want to learn the solution map $F^\dagger$ which maps each pair $(s, b_s)$ to a function $u_s^\dagger : M_s \to \mathbb{R}^{d_u}$ that solves the trace of the PDE system on that particular surface geometry, where $b_s : M_s \to \mathbb{R}^{d_b}$ represents boundary data or physical parameters on the surface.

A classic example of this problem is the study of fluid flow around a moving object. The governing partial differential equations (PDEs) are the Navier-Stokes equations, with the object's shape, $s$, representing the geometric domain. Of particular interest are surface quantities like pressure and wall-shear stress, which are integrated over the object's surface to calculate the aerodynamic drag. By developing an efficient mapping from the object's shape and other geometric parameters to these surface fields, it becomes possible to directly minimise drag. This optimisation problem has been extensively studied, for example by Wei et al. (2023) and Abbas et al. (2023).

The challenge in this setting, unlike traditional supervised learning on Euclidean spaces, is that the domain of $u_s^\dagger$ changes with each shape $M_s$. Because shapes in the dataset can vary in both geometry and topology, traditional neural networks—which require fixed input/output dimensions and cannot account for the geometric structure of each shape—are not directly applicable.

**Supervised learning problem.** In practice, each surface $M_s$ is discretised as a triangular mesh $M_s^h = (\mathcal{V}_s, \mathfrak{F}_s, X_s)$ with $n_s = |\mathcal{V}_s|$ vertices having positions $X_s \in \mathbb{R}^{n_s \times 3}$ and face connectivity $\mathfrak{F}_s$ defining the triangulation. Functions on the surface become vectors at vertices, with inputs $b_s \in \mathbb{R}^{n_s \times d_b}$ representing boundary data and outputs $u_s \in \mathbb{R}^{n_s \times d_u}$ representing the solution field. While the continuous operator $F^\dagger$ maps between infinite-dimensional spaces, we learn from a discrete ground truth map $F^h : (M_s^h, b_s) \mapsto u_s$ obtained from numerical simulations. Henceforth, we will use the symbol $y_s := u_s$ to define the solution field to make it clear that this quantity is the target data. Given dataset $\mathcal{D} = \{(M_{s_i}^h, b_{s_i}, y_{s_i})\}_{i=1}^N$, we seek a neural network that processes functions on arbitrary meshes while respecting geometric structure and handling varying mesh sizes and topologies.

### 2.2 DIFFUSIONNET

DiffusionNet, as proposed by Sharp & Crane (2020), employs heat diffusion as a learnable convolution for surface analysis. It discretises the heat operator's action, $e^{\Delta_M t} : v(t) = e^{\Delta_M t} v(0)$ on a continuous signal $v(0)$ on an embedded surface, $M$. Here $\Delta_M$ represents the Laplace-Beltrami operator (LBO). The diffusion time, $t$, is a critical parameter that controls the scale of the convolution: a small $t$ captures local feature dependencies, while a large $t$ facilitates global message-passing.

On a mesh $M_s^h$, the LBO is approximated by the *lumped cotangent Laplacian* $L_s$. Its first $K$ eigenvectors and eigenvalues are denoted by $\Phi_s \in \mathbb{R}^{n_s \times K}$, $\Lambda_s = \text{diag}(\lambda_1, \ldots, \lambda_K)$, respectively.

The *mass matrix* $A_s$, derived from $L_s$ is used to approximate the continuous $L^2(M_s)$ inner product. We refer to Sharp & Crane (2020) for details on how to compute these quantities.

For features $v \in \mathbb{R}^{n_s \times d_c}$ with channel dimension $d_c$, and learned diffusion times $t = (t_1, \ldots, t_{d_c})$, the diffusion operator is built by applying the heat operator, described above in continuum, channelwise:

$$\mathcal{P}_t(v) = \begin{bmatrix} \Phi_s \exp(-\Lambda_s t_1)\Phi_s^T A_s v_1 & \cdots & \Phi_s \exp(-\Lambda_s t_{d_c})\Phi_s^T A_s v_{d_c} \end{bmatrix} \tag{2}$$

where $v_j$ is the $j$-the column of $v$ and $\exp(-\Lambda_s t_j) = \text{diag}(e^{-\lambda_1 t_j}, \ldots, e^{-\lambda_K t_j})$. This creates a learned spectral filter bank where each channel captures information at a different geometric scale, analogous to multi-scale convolutions in traditional CNNs but adapted to the non-Euclidean setting.

While the diffusion operator efficiently propagates feature information across the mesh, its inherently isotropic nature limits its effectiveness. To address this, DiffusionNet applies a learnable gradient operator $\Gamma_{\theta_g}(\cdot)$ to capture directional information from $\mathcal{P}_t(v)$. Since this operator is not a primary focus of our work, we refer the reader to Sharp et al. (2022) for a detailed discussion of how the gradients features are extracted. Finally, a pointwise MLP, $\mathcal{F}_w$, is applied to the concatenated features $[v \mid \mathcal{P}_t(v) \mid \Gamma_{\theta_g}(\mathcal{P}_t(v))]$. The combination of these three elements results in a single DiffusionNet block $\mathcal{B}_\theta$ with parameters $\theta = (t, \theta_g, w)$ given by

$$\mathcal{B}_\theta(v; M_s^h) := v + \mathcal{F}_w\left([v \mid \mathcal{P}_t(v) \mid \Gamma_{\theta_g}(\mathcal{P}_t(v))]\right) \tag{3}$$

The inclusion of $M_s^h$ as a second input signifies that the block's operation is dependent on the geometric quantities required by the diffusion and gradient operators such as $\Phi_s, \Lambda_s, A_s$.

The full DiffusionNet architecture is a composition of $L$ such blocks, along with pointwise lifting and projection MLPs $\mathcal{L}_{w_{\text{lift}}} : \mathbb{R}^{n_s \times d_{\text{in}}} \to \mathbb{R}^{n_s \times d_c}$ and $\mathcal{Q}_{w_{\text{proj}}} : \mathbb{R}^{n_s \times d_c} \to \mathbb{R}^{n_s \times d_{\text{out}}}$. The complete map is defined as

$$D_\theta(v; M_s^h) := \mathcal{Q}_{w_{\text{proj}}} \circ \mathcal{B}_{\theta_L} \circ \cdots \circ \mathcal{B}_{\theta_1} \circ \mathcal{L}_{w_{\text{lift}}}(v). \tag{4}$$

DiffusionNet's effectiveness stems from its intrinsic operations, which adapt to the unique geometry of each surface. The spectral basis allows for multi-resolution processing while maintaining resolution invariance, i.e. models trained on one mesh resolution generalize to others.

To adapt DiffusionNet for physics applications, we augment the input field with various geometric and physical features such as: boundary conditions for the PDE (1), vertex coordinates and normals, heat kernel signatures (Sharp et al., 2022), and any relevant physical parameters. The specific features used in our experiments are detailed in Section 4.

**Connection with Neural Operators** Since the key components of DiffusionNet–diffusion and gradients–can be formulated in continuum, the architecture can learn PDE operators on complex geometries while maintaining geometric consistency and resolution invariance. This capability is similar to that of a Neural Operator. We discuss this connection in detail in Appendix A.

## 3 METHODOLOGY

Learning from finite training data inherently introduces uncertainty—we cannot perfectly recover $F^h$ across all possible geometries, and our confidence should vary on unseen data. We therefore extend DiffusionNet to learn a distribution over possible mappings rather than a single approximation, enabling uncertainty quantification for predictions on novel geometries.

### 3.1 PROBABILISTIC DIFFUSIONNET

Our goal is to create a probabilistic model for surface fields on a mesh by leveraging the inductive biases of a geometric model architecture. To achieve this, we reformulate DiffusionNet's core mechanism—the deterministic diffusion process—into a stochastic one. This approach injects a noising mechanism directly into the underlying convolution operation, leading to a set of probabilistic operations that are both intuitive and interpretable. To define these operations, we first obtain a spectral representation of the solution to the *stochastic heat equation* in Theorem 3.1. We then use this representation to build a stochastic diffusion operator for carrying out a stochastic convolution (message-passing) layer that becomes the building block of a probabilistic model for surface fields on a mesh.

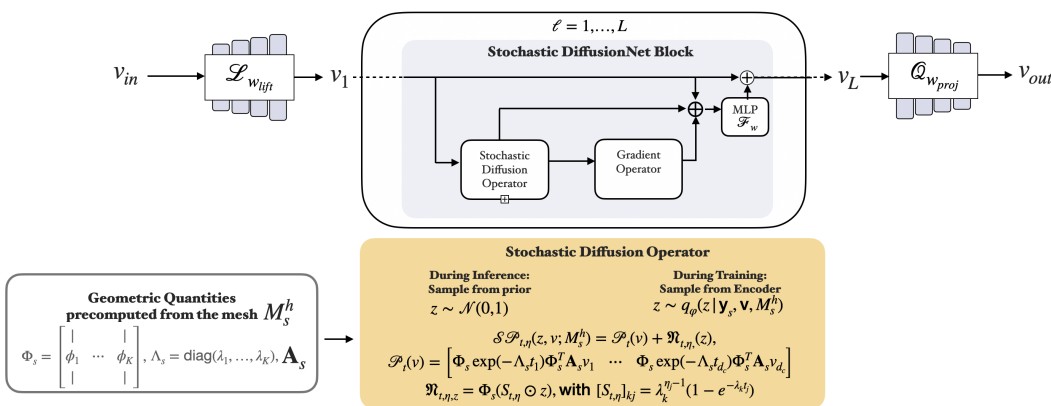

Figure 1: Architecture of the Stochastic DiffusionNet Block. Note that except the Stochastic Diffusion Operator, all other operations remain the same as DiffusionNet (Sharp et al., 2022).

A stochastic process, on an embedded surface $M$ with LBO $\Delta_M$, can be obtained as a randomly perturbed heat equation through the addition of spatial (coloured) noise:

$$\partial_t v(t) = \Delta_M v(t) + \mathcal{W}^Q, \quad v(0) = v_0 \tag{5}$$

where the spatial noise $\mathcal{W}^Q$ is an $L^2(M)$-valued Gaussian random field such that its covariance operator $Q$ has LBO eigenvectors $\{\phi_k\}_{k=0}^\infty$ and eigenvalues $\{q_k\}_{k=0}^\infty$ (see, Prévôt & Röckner (2007)[Proposition 2.1.6] and Lototsky & Rozovskii (2009)[Thm. 3.2.15]). The following theorem characterizes the solution of (5). A more complete statement and proof is provided in Appendix B, where we also provide an upper bound on the expected squared norm of $v(t)$ under suitable initial conditions and choice of $\{\lambda_k, z_k\}_{k=0}^\infty$.

**Theorem 3.1.** *There exists a unique strong solution of* (5) *with the spectral representation:*

$$v(t) = \sum_{k=0}^\infty \left( e^{-\lambda_k t} \langle v_0, \phi_k \rangle_{L^2(M)} + \frac{\sqrt{q_k}}{\lambda_k} (1 - e^{-\lambda_k t}) z_k \right) \phi_k , \tag{6}$$

*where $\{z_k\}_{k=0}^\infty \overset{\text{i.i.d.}}{\sim} \mathcal{N}(0,1)$. Moreover, conditional on $v_0$, $v(t)$ is an $L^2(M)$-valued Gaussian random field.*

Building on this foundation, we now construct our stochastic variant of the diffusion operator $\mathcal{P}_t$. To do this, we first parameterise the noise covariance via $q_k = \lambda_k^{2\eta}$ where $\eta$ controls the noise spectrum (see section G.5 for further details). This parameterisation avoids basis ambiguities as discussed in Lim et al. (2022); Zhang et al. (2024) (see Section A.1). For diffusion times $t = (t_1, \ldots, t_{d_c})$, noise parameters $\eta = (\eta_1, \ldots, \eta_{d_c})$ and random matrix $z \in \mathbb{R}^{K \times d_c}$ with $z_{kj} \sim \mathcal{N}(0,1)$, we can extend the diffusion operator $\mathcal{P}_t$ to a corresponding stochastic one by discretising Eq. (18) on the mesh $M_s^h$:

$$\mathcal{SP}_{t,\eta}(z, v; M_s^h) = \mathcal{P}_t(v) + \mathfrak{N}_{t,\eta,}(z), \tag{7}$$

where the noise perturbation term $\mathfrak{N}_{t,\eta,z} = \Phi_s(S_{t,\eta} \odot z)$, with $[S_{t,\eta}]_{kj} = \lambda_k^{\eta_j - 1}(1 - e^{-\lambda_k t_j})$, and $\odot$ denotes element-wise multiplication. In practice, we repeat the random variable across channels, which reduces $z$ to a $K$-dimensional random vector. A Stochastic DiffusionNet block (see Figure 1) $\mathcal{SB}_\theta$ with parameters $\theta = (t, \eta, \theta_g, w)$ is then given by:

$$\mathcal{SB}_\theta(z, v; M_s^h) := v + \mathcal{F}_w \left( [v \mid \mathcal{SP}_{t,\eta}(z, v) \mid \Gamma_{\theta_g}(\mathcal{SP}_{t,\eta}(z, v))] \right). \tag{8}$$

Probabilistic DiffusionNet (PDN), defines a hierarchical generative model encapsulated by the following generative process:

$$z_1, \ldots, z_L \sim \mathcal{N}(0, I_K)$$
$$SD_\theta(z_{1:L}, v; M_s^h) := \mathcal{Q}_{w_{\text{proj}}} \circ \mathcal{SB}_{\theta_L} \circ \cdots \circ \mathcal{SB}_{\theta_1} \circ \mathcal{L}_{w_{\text{lift}}}(z_{1:L}, v) \tag{9}$$

where $z_{1:L} = \{z_1, \ldots, z_L\}$ and $\theta = \{w_{\text{lift}}, \theta_1, \ldots, \theta_L, w_{\text{proj}}\}$.

The generative process above lets us define a marginal distribution:

$$p_{\boldsymbol{\theta}}(\text{u}|\text{v}_{\text{input}}, M_s^h) = \int \delta(\text{u} - \text{SD}_\theta(z_{1:L}, v; M_s^h)) \prod_{\ell=1}^{L} p(z_\ell) dz_{1:L} \tag{10}$$

of a surface field u on a mesh $M_s^h$, where $\delta(\cdot)$ denotes a Dirac delta distribution. The hierarchical structure induces uncertainty at multiple geometric scales, with each layer $\ell$ adding stochastic perturbations at scale $t_\ell$ and magnitude $\eta_\ell$.

### 3.2 Variational Inference

Having defined a probabilistic generative model for fields on a mesh surface, we can use it to model the PDE solution fields obtained from simulation. Thus, we proceed to learn the parameters $\theta$ given the training dataset $\mathcal{D}$.

**Observational model.** We consider an observation model for the simulated solution field $\text{u}_\text{s}$ given by the following measurement equation:

$$\text{y}_\text{s} = \text{u} + \boldsymbol{\epsilon}, \tag{11}$$

where $\text{u} \sim \text{p}_\theta(\text{u}|\text{v}_{\text{input}}, M_s^h)$ (10) is the probabilistic surrogate model, and $\boldsymbol{\epsilon} \sim \mathcal{N}(0, \sigma^2 \mathbb{I}_{n_{s_i}})$ is a measurement noise. This measurement noise is added to capture some form of model discrepancy and to stabilise the training dynamics.

**ELBO.** Our probabilistic model is a parametrised latent-variable model, where $z$ represents the random latent variable. Our goal is to maximize the marginal log-likelihood. After collecting the solution fields, inputs, latent variables and meshes corresponding to all shapes in $\text{Y} = \{\text{y}_{\text{s}_i}\}_{i=1}^{N}$, $\text{V} = \{\text{v}_{\text{input}_i}\}_{i=1}^{N}$, $\mathcal{M} = \{M_{s_i}^h\}_{i=1}^{N}$, $Z = \{z^i\}_{i=1}^{N}$, we can write the marginal log-likelihood as:

$$\ln p_{\theta,\sigma}(\text{Y}|\text{V}, \mathcal{M}) = \ln \int p_{\theta,\sigma}(\text{Y}|\text{V}, \mathcal{M}, Z) p(Z) dZ, \tag{12}$$

$$p_{\theta,\sigma}(\text{Y}|\text{V}, \mathcal{M}, Z) = \prod_{i=1}^{N} \mathcal{N}(\text{y}_{\text{s}_i}; \text{SD}_{\boldsymbol{\theta}}(z_{1:L}^{i}, \text{v}_i), \sigma^2). \tag{13}$$

Since the integral cannot be computed analytically due to the non-linear relationship between $\text{y}_\text{s}$ and $z$, we employ *variational inference* (Murphy, 2022). We introduce a family of distribution $q_\varphi(Z|\text{Y}, \text{V}, \mathcal{M})$, parametrised by $\varphi$, to approximate the posterior $p(Z|\mathcal{D})$. We optimize $(\theta, \sigma, \varphi)$ by maximizing the evidence lower bound (ELBO):

$$\mathcal{L}(\theta, \varphi) = \mathbb{E}_{q_\varphi}[p_{\theta,\sigma}(\text{Y}|\text{V}, \mathcal{M}, Z)] - \mathcal{D}_{KL}(q_\varphi(Z|\text{Y}, \text{V}, \mathcal{M}) \| p(Z)) \le \ln p_{\theta,\sigma}(\text{Y}|\text{V}, \mathcal{M}). \tag{14}$$

**Amortized encoder.** We adopt an amortized approach for the variational approximation given by:

$$q_\varphi(Z|\text{Y}, \text{V}, \mathcal{M}) = \prod_{i=1}^{N} \prod_{l=1}^{L} \mathcal{N}(z_\ell^i; \mu_\ell^\varphi(\text{y}_{\text{s}_i}, \text{v}_{\text{input}_i}, M_{s_i}^h), \Sigma_l^\varphi(\text{y}_{\text{s}_i}, \text{v}_{\text{input}_i}, M_{s_i}^h)) \tag{15}$$

where $\mu_\ell^\varphi$ and $\Sigma_\ell^\varphi$ are simple variants of DiffusionNet with an aggregation layer (see Section C for details). We use the *reparameterisation-trick* to obtain a Monte Carlo estimate of the gradient of the ELBO, to carry out its maximisation using a stochastic gradient descent algorithm.

**Uncertainty quantification** Once we have learned the optimal parameters $(\theta_\star, \sigma_\star, \varphi_\star)$ by maximizing (14), we may sample from the observational predictive distribution, for unseen shape $s_\star$:

$$p_{\theta,\sigma}(\text{y}_{\text{s}_\star}|\text{v}_{\text{input}_\star}, M_{s_\star}) = \int \text{p}_{\theta,\sigma}(\text{y}_{\text{s}_\star}|\text{v}_{\text{input}_\star}, M_{s_\star}^h, z^\star) p(z^\star) dz^\star. \tag{16}$$

The variance of the predictive distribution (16) reflects the model's noise sensitivity, with shape-dependent modulation through $q_k(s) = \lambda_k(s)^{2\eta}$ ensuring higher variance for unusual geometric

324 spectra. While this correlates with epistemic uncertainty patterns (higher on novel shapes), it is
325 technically learned *heteroscedastic aleatoric uncertainty* since parameters are fixed post-training.
326 True epistemic uncertainty would require treating $\theta, \sigma$ as random variables. Nevertheless, PDN's
327 geometrically-structured predictive variance provides valuable confidence estimates, effectively
328 warning when predictions extrapolate beyond the training distribution.

## 4 EXPERIMENTAL RESULTS

We benchmark our approach in two phases: first evaluating the accuracy of Probabilistic DiffusionNet
(PDN) against state-of-the-art (SOTA) geometric deep learning models for PDE surrogacy, then
comparing PDN against baseline UQ methods to assess predictive accuracy and uncertainty quality.

**Datasets.** We evaluate on two standard computational fluid dynamics datasets, both containing
Reynolds-Averaged Navier-Stokes simulations on varying geometries: **ShapeNet car** (Umetani &
Bickel, 2018), consisting of car meshes with 3,856 vertices, and **Ahmed bodies** (Li et al., 2023d),
consisting of simplified automotive test shapes with meshes having $\sim 10^5$ vertices per shape.
Following Li et al. (2023d), we apply the same preprocessing and test/train splits, learning a mapping
from shape geometry to surface pressure distribution.

**Baselines and Metrics.** In Section 4.1, we compare PDN's accuracy against three categories of
surrogate models: i) Neural operators that handle mesh data, including GNO, GINO, and Fengbo
(Pepe et al., 2025); ii) Transformer-based approaches like Transolver (Wu et al., 2024) and Point
Cloud Transformer (PCT); and iii) Graph-based methods, including GraphSage (Hamilton et al.,
2017), MeshGraphNet (Pfaff et al., 2020), and the original DiffusionNet (DN). We evaluate predictive
accuracy using the relative $L^2$ loss. We left out benchmarking some of the graph and transformer-
based models for Ahmed bodies, due to the large mesh size. We report PDN's performance for
an extended set of ShapeNet car dataset in Section G.1. In Section 4.2, we evaluate both accuracy
using RMSE, due to our probabilistic output, and uncertainty quality using total miscalibrated area
(MCAL), negative predictive log-likelihood (NLL), and interval score (IS). All metrics are computed
using the uncertainty toolbox package Chung et al. (2021). Note that there is no single metric that
captures every aspect of probabilistic predictions, so we resort to the collection of metrics as these
are the most commonly used and there is a rich literature to support them. We discuss further details
of the metrics and their choices in Section E. We compare PDN against GINO and Transolver, each
enhanced with standard UQ methods: Monte Carlo Dropout (DO), Laplace Approximation (LA), and
Model Ensembling (ME), with 10 models. As the code for the conformal prediction method proposed
in Ma et al. (2024b) has not been released, we do not provide a comparison to it. In Section F we
discuss the motivation behind choosing these UQ methods and their implementation. Finally in
Section 4.3 we evaluate PDN's prediction quality on *out-of-distribution* (OOD) shapes by creating a
different split of the **ShapeNet car** dataset based on geometric similarity.

**Implementation.** For all experiments we used 16 blocks and chose $K = 128$ modes, based on
ablations (see Section G.2). To obtain the cotan-Laplacian, we used the robust Laplacian method
proposed in Sharp & Crane (2020) and used the approach in Sharp et al. (2022) to approximate the
gradient features. Note that these were obtained beforehand and cached. For the input field $v_{\text{input}}$, we
simply used the vertex positions $X_s$. In the Ahmed body dataset, the inlet velocity $\rho$ changes between
samples, so we repeated this scalar value across vertices to form the input $v_{\text{input}} = [X_s | \rho 1_{n_s}]$, where
$1_{n_s} \in \mathbb{R}^{n_s}$ is a vector of ones. We discuss further details of implementation including optimisation in
Section D. We used `PyTorch` (Paszke et al., 2017) for implementation. Experiments ran on a single
NVIDIA A100 GPU. The code is available at `retracted`.

### 4.1 PREDICTIVE ACCURACY

We compare the predictive performance of PDN against baseline methods across both datasets
(Table 1). Our results demonstrate that both DiffusionNet (DN) and PDN achieve SOTA performance
on the ShapeNet car dataset and outperform all other implementations on the larger Ahmed bodies
dataset. These results validate that shape-dependent operators provide an effective geometric inductive
bias for PDE surrogate modelling. Notably, PDN's performance closely matches that of the original
DN implementation while adding UQ capabilities, and incurring a lower computational cost with a
smaller parameter count, (see Table 2 and Section G.3). The minimal impact on predictive accuracy

Table 1: Performance of various methods for predicting the pressure field on a surface, evaluated using the relative $L^2$ (RL2) error. The values, shown in units of $10^{-2}$, represent the average RL2 error across all samples in the test dataset, with lower values indicating higher accuracy. For PDN we used the mean of 100 samples from the predictive distribution (16).

| The ShapeNet car dataset | | The Ahmed bodies dataset | |
|---|---|---|---|
| Methods | ↓RL2 | Methods | ↓RL2 |
| PointNet (Charles et al., 2017a) | 11 | MeshGraphNet (Pfaff et al., 2020) | 13.9 |
| GraphSage (Hamilton et al., 2017) | 10.5 | UNet (with interpolation) | 11.2 |
| GNO (Li et al., 2020a) | 18.8 | FNO (with interpolation) | 12.6 |
| GeoFNO (Li et al., 2023b) | 15.9 | GINO (Li et al., 2023d) | 8.31 |
| GINO (Li et al., 2023d) | 7.1 | Fengbo (Pepe et al., 2025) | 10.7 |
| Fengbo (Pepe et al., 2025) | 8.9 | Transolver (Wu et al., 2024) | 7.1 |
| PCT (Guo et al., 2021) | 7.7 | DN | **5.4** |
| Transolver (Wu et al., 2024) | **6.2** | PDN | 6.2 |
| DN | 6.3 | | |
| PDN | 6.3 | | |

when moving from DN to PDN, despite incorporating probabilistic elements, highlights the advantage of capturing uncertainty through the message-passing operation.

## 4.2 UNCERTAINTY QUANTIFICATION

In Table 2 we compare the quality of UQ. We evaluate models that create probabilistic predictions of scalar pressure fields for new shapes. Each model learns a distribution over possible pressure fields, allowing us to quantify its uncertainty by sampling candidate predictions. These samples are then used to calculate uncertainty quantification metrics, assessing the model's concentration, calibration, and coverage.

We furnished the average values of metrics across test samples. Moreover, excluding ME, we carried out additional repeats of the experiments and averaged the metrics across these. Since each metric captures only a certain aspect of the UQ, it is important that a method performs in balanced way across all these metrics. To highlight this aspect we ranked the performance of each of the competing method on individual metrics for each dataset. We then averaged these ranks across all the metrics and datasets. We noticed, based on the average rank, that TS-ME followed by PDN had the best average rank, indicating their dominance in terms of producing a balanced performance. We also noticed that for some methods, LA, DO, the performance can vary drastically across datasets and models. Notice how the PDN metrics are always well balanced across the datasets. This shows that PDN can reliably and robustly produce predictive uncertainties in a problem-agnostic manner. Fig. 2

Table 2: UQ metrics for surface pressure field prediction . Each metric is first averaged across the test set shapes, and then averaged across three repeats of the experiments. We also report the average rank (Rank) of each method across all metrics (a lowest value indicates most balanced performance considering all metrics)

| Methods | Rank↓ | ShapeNet car | | | | Ahmed bodies | | | |
|---|---|---|---|---|---|---|---|---|---|
| | | RMSE↓ | NLL↓ | MCAL↓ | IS↓ | RMSE↓ | NLL↓ | MCAL↓ | IS↓ |
| GINO-DO | 6.50 | 6.59 | 2.44 | **0.07** | 10.01 | 32.74 | 5.28 | 0.24 | 121.83 |
| GINO-LA | 8.25 | 4.27 | 29.30 | 0.23 | 14.81 | 34.58 | 5.27 | 0.24 | 110.31 |
| GINO-ME | 5.38 | 3.87 | **2.21** | 0.13 | 6.35 | 28.82 | 4.65 | 0.29 | 126.69 |
| TS-DO | 5.06 | **3.86** | 2.47 | 0.11 | 9.57 | 21.17 | 4.62 | 0.30 | 119.55 |
| TS-LA | 6.13 | 4.02 | 5.84 | 0.09 | 11.19 | 25.01 | 5.29 | 0.26 | 100.66 |
| TS-ME | **2.81** | 4.06 | 2.31 | 0.11 | **5.66** | 24.95 | 3.26 | **0.08** | **37.15** |
| DN-DO | 8.25 | 4.61 | 2.76 | 0.21 | 20.55 | 26.30 | 4.92 | 0.30 | 190.18 |
| DN-LA | 5.25 | 4.18 | 7.52 | 0.09 | 12.32 | 20.79 | 4.72 | 0.23 | 67.01 |
| DN-ME | 4.13 | 5.11 | 2.36 | 0.1 | 8.8 | 27.35 | **3.04** | 0.1 | 50.50 |
| PDN (Ours) | 3.25 | 3.91 | 2.77 | 0.08 | 7.99 | **18.28** | 4.23 | 0.23 | 67.33 |

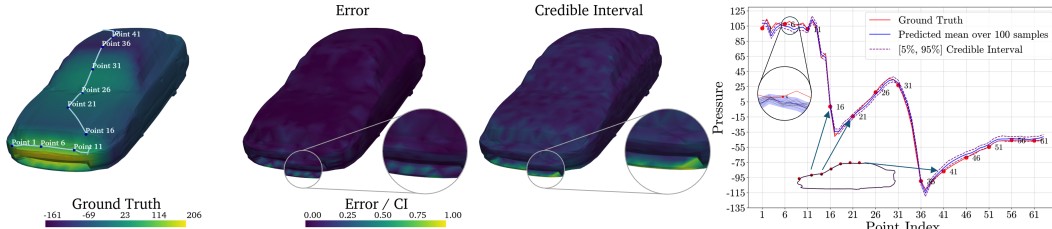

Figure 2: **Visualization of uncertainty estimates produced by PDN for a Shape-Net car**. a) Ground-truth pressure field and one sampled geodesic path of vertices from the front to the back of the car. b) Normalized absolute error between the predicted mean and the ground-truth field. c) Normalized Credible Range, i.e., difference between the 5th and 95th percentiles, of 100 samples from the predictive distribution, which correlates well, see the inset region, with the absolute error. d) Comparison of the predicted samples and ground-truth along the geodesic path points shown in a).

Table 3: UQ metrics for surface pressure field prediction (test set averages) on OOD shapes.

| Methods | ShapeNet car OOD | | | |
| --- | --- | --- | --- | --- |
| | RMSE↓ | NLL↓ | MCAL↓ | IS↓ |
| GINO-DO | 7.65 | 5.67 | **0.06** | 13.85 |
| TS-DO | 5.99 | 3.45 | 0.30 | 42.74 |
| DN-DO | 5.68 | **2.77** | 0.12 | 16.57 |
| PDN (Ours) | **5.54** | 3.50 | 0.09 | **10.22** |

shows how PDN's uncertainty estimate correlates well with the prediction error for a ShapeNet car sample. Whilst model ensembling yields the strongest UQ performance, when combined with TS, it comes with the significant cost of training multiple (in our case 10) models on different subsets of the data, and so is infeasible for industrial scale applications. By contrast, PDN yields good UQ with no additional cost.

### 4.3 OUT-OF-DISTRIBUTION PREDICTION

For this experiment, we clustered the ShapeNet car dataset using spectral clustering (Shi & Malik, 2000), where we used a pairwise chamfer distance to measure similarity. This lead to discovering two clusters having 471 (used as training set) and 140 (used as test set) shapes respectively. We then compared the performance of PDN, GINO-DO, TS-DO and DN-DO (since these do not incur additional compute cost such as LA and ME methods) on this dataset using the same metrics used for previous UQ experiments. We furnish the UQ metrics in Table 3 and visualisations in Fig. 3. The above results reinforce PDN's strengths. All methods, except GINO-DO, show similar accuracy, with PDN having a slight edge. Crucially, PDN's coverage (IS) is significantly better, indicating it better handles distribution shifts. While GINO-DO shows good calibration, this is a result of overly wide prediction intervals, which negatively impacts its IS score.

## 5 CONCLUSIONS

DiffusionNet (DN) is an efficient graph neural network that uses a diffusion mechanism for message passing. We introduced Probabilistic DiffusionNet, an extension that introduces stochasticity within this message-passing operation of DiffusionNet by modifying the diffusion mechanism, yielding spatially correlated predictive distributions. This approach delivers superior uncertainty quantification on standard benchmarks while requiring substantially fewer parameters and computational resources than other popular alternatives.

For the DN/PDN architecture, one has to eigendecompose the cotan-Laplacian; despite efficient implementations, this can become a problem for large meshes. Future work may consider approximations to the eigendecomposition for efficiency. Future work may also consider applying the model to real-world applications requiring UQ such as design optimisation and active learning.

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

# Probabilistic DiffusionNet: A geometry informed probabilistic generative neural operator. Supplementary material

## A ADDITIONAL DETAILS ABOUT DIFFUSIONNET

### A.1 CONNECTION WITH NORM

In Chen et al. (2024), the authors generalise Fourier Neural Operators (FNOs) to an arbitrary Riemannian manifolds $(M, g)$ by replacing the Fourier transform with projection onto the LBO basis $\{\phi_k\}$. The integral transformation layer of Chen et al. (2024) $\mathcal{K}_{\mathcal{T}} \in C(L^2(M)^{d_c}; L^2(M)^{d_c})$ is defined by

$$[\mathcal{K}_{\mathcal{T}}(M)(v)]_c = \sum_{k=0}^{\infty} \sum_{r=1}^{d_c} \mathcal{T}_k^{cr} \langle \phi_k, v_r \rangle_{L^2(M)} \phi_k, \quad c \in [d_c], \tag{17}$$

where $\mathcal{T} \in \ell^{\infty}(\mathbb{N}_0)^{c \times c}$ is a bounded multiplier (in practice, one restricts to $\mathcal{T}$ such that $\mathcal{T}_k = 0$ for all $k \geq K$). The resulting architecture with lifting, projection, and residual skips is called Neural Operator on Riemannian Manifolds (NORM). The projection on to the LBO basis only corresponds to a Fourier transform in the strict sense if the manifold $M$ is a locally compact abelian group Rudin (2017).

In special cases, we recover familiar neural operators:

- If $M = \mathbb{T}^d = [0, 2\pi]^d / \sim$ is the flat torus, then $\mathcal{K}_{\mathcal{T}}(M)$ becomes the convolution layer in FNOs since the LBO basis is given by $\{\phi_k\} = \{e^{i\langle k, \cdot \rangle}\}_{k \in \mathbb{Z}^d}$ Zelditch (2017)[Sec. 4.3],
- If $M = S^2$ is the 2-sphere, then $\mathcal{K}_{\mathcal{T}}(M)$ is the convolution operator in Spherical Neural Operators Bonev et al. (2023) since the LBO basis $\{\phi_k\}$ consists of the spherical harmonics Zelditch (2017)[Sec. 4.4].

The diffusion operator $\mathcal{P}_t$ introduced in Section 2.2, in continuum, is a special case of (17) with spectral multiplier

$$\mathcal{T}_k^{cr} = e^{-\lambda_k t_r} \delta_{cr},$$

where $\delta_{cr} = 0$ if $c \neq r$ and 1 otherwise.

While NORM provides a general framework for neural operators on manifolds, our primary focus is inductive learning across different shapes. This introduces an important consideration: eigenfunctions in a given eigenspace are defined only up to orthogonal transformations—particularly, when the eigenvalue is simple (multiplicity one), they are defined only up to a sign flip. This phenomenon creates the "basis ambiguity problem" well-documented in spectral graph neural networks Huang et al. (2023b); Lim et al. (2022); Zhang et al. (2024).

To address ambiguities when learning across shapes, researchers have proposed several strategies:

- Use a restricted class of multipliers like $\mathcal{P}_t$ which are invariant to sign flips, and more general, orthogonal transformations (see Huang et al. (2023b); Lim et al. (2022); Zhang et al. (2024)),
- Apply an eigenfunction canonisation procedure Ma et al. (2023; 2024a), or
- Randomly apply an orthonormal transformation to eigenfunctions in the same eigenspace (effectively flipping signs in the simple eigenvalue case) Dwivedi et al. (2023)[App. E.1].

Therefore, the general spectral multiplier parameterisation in (17) requires additional considerations to be suitable for inductive learning across shapes. For this reason, we do not benchmark against this architecture.

DiffusionNet uses the restricted diffusion operator $\mathcal{P}_t$, which is naturally immune to the basis ambiguity issue. Our experimental results demonstrate that the restricted nature of $\mathcal{P}_t$ does not compromise performance in practice, especially when complemented with the anisotropic gradient operator, which provides additional discriminative power.

## A.2 CONNECTION WITH GINO

In Li et al. (2023d)[Sec. 2], the authors introduce a problem setting analogous to ours (cf. Section 2.1), addressing the challenge of learning the solution map of a boundary value problem from geometry to solution. Their approach, Geometry-Informed Neural Operator (GINO), consists of three components: i) a graph neural operator (GNO) encoder Li et al. (2020a), ii) an FNO latent on a rectangular grid, and iii) a GNO decoder. A similar strategy was employed in Li et al. (2023c), albeit with different encoders and decoders that are diffeomorphic mappings between manifolds. The advantage of moving to a rectangular grid is to leverage the speed of fast Fourier transforms to perform global convolutions.

The encoder of GINO maps a function on a general manifold to a function on a rectangular latent domain, and the decoder performs the opposite operation (though not the true inverse of the encoder). Given embedded manifolds $\mathcal{D}_1 \subset \mathbb{R}^d$ and $\mathcal{D}_2 \subset \mathbb{R}^d$ and function spaces $\mathcal{F}_1(\mathcal{D}_1)^{d_1}$ and $\mathcal{F}_2(\mathcal{D}_2)^{d_2}$, a local linear GNO layer $\mathcal{G}_\theta : \mathcal{F}_1(\mathcal{D}_1)^{d_1} \to \mathcal{F}_2(\mathcal{D}_2)^{d_2}$ is defined by:

$$\mathcal{G}_\theta(v)(x) = \int_{B_r^{(d)}(x) \cap \mathcal{D}_1} k_\theta(x, y) v(y) \mu_{\mathcal{D}_1}(dy), \quad x \in \mathcal{D}_2,$$

where $B_r^{(d)}(x) = \{y \in \mathbb{R}^d : |x - y|_{\mathbb{R}^d} < r\}$ is a ball of radius $r \in (0, \infty]$ in $\mathbb{R}^d$, $k_\theta \in C(\mathbb{R}^d \times \mathbb{R}^d; \mathbb{R}^{d_2 \times d_1})$ is a parameterised matrix of kernels, and $\mu_{\mathcal{D}_1}$ is a measure on $\mathcal{D}_1$.

A crucial feature of GINO is that the kernel $k_\theta$ is defined on the entire embedding space $\mathbb{R}^d$, which allows the decoder to be interpreted as mapping to functions defined throughout $\mathbb{R}^d$. This design makes GINO particularly well-suited to learn the extended solution map discussed in Li et al. (2023d)[Sec. 2]. The trace solution map $F^\dagger$ is not explicitly discussed in Li et al. (2023d). While the experiments in Li et al. (2023d) (ShapeNet car and Ahmed body) focused entirely on surface predictions—effectively learning the trace solution map to functions on Lebesgue measure-zero subsets of the embedding space $\mathbb{R}^3$—the ability to decode to functions throughout the entire volume has been recognised and utilised in subsequent works (see, e.g., Table 3 in Wu et al. (2024)).

In contrast to GINO, DiffusionNet and Probabilistic DiffusionNet are inherently constrained to the surface for every shape input $s \in \mathcal{S}$. We believe this architectural constraint introduces a strong geometric inductive bias, enabling DiffusionNet to achieve competitive performance with a significantly smaller parameter count than GINO while focusing precisely on the surface of interest for the trace solution map $F^\dagger$. By working directly on the manifold rather than extending to the ambient space, DiffusionNet exploits the intrinsic geometry of the problem.

## B PROOF OF THEOREM 3.1

Let $(M, g)$ denote a smooth Riemannian manifold embedded in $\mathbb{R}^3$ with the associated pullback embedded metric $g$. Denote by $W^{\mathfrak{s},2}(M)$, $\mathfrak{s} \in \mathbb{R}$, the corresponding scale of $L^2$-Sobolev spaces. We note that if $\mathfrak{s} > 1$, then $W^{\mathfrak{s},2}(M) \subset C(M)$ Behzadan & Holst (2022)[Thm. 9.14]. Let $(\Omega, \mathcal{F}, \mathbb{P})$ be a complete probability space supporting an $L^2(M)$-valued Gaussian random field $\mathcal{W}^Q : \Omega \times M \to \mathbb{R}$ is such that its covariance operator $Q \in \mathcal{L}(L^2(M), L^2(M))$ has LBO eigenvectors $\{\phi_k\}_{k=0}^\infty$ and eigenvalues $\{q_k\}_{k=0}^\infty$ (see, Prévôt & Röckner (2007)[Proposition 2.1.6] and Lototsky & Rozovskii (2009)[Thm. 3.2.15]).

**Definition B.1.** A measurable stochastic process $v : \Omega \times [0, T] \to L^2(M)$ is called a weak solution of (5) on the interval $[0, T]$ if $\mathbb{P}$-a.s., $v \in C([0, T]; L^2(M))$ and $\mathbb{P}$-a.s., for all $t \in [0, T]$,

$$\langle v(t), \phi \rangle_{L^2(M)} = \langle v_0, \phi \rangle_{L^2(M)} + \int_0^t \langle v(s), \Delta_M \phi \rangle_{L^2(M)} ds + \langle \mathcal{W}^Q, \phi \rangle_{L^2(M)} t.$$

If, in addition, $\mathbb{P}$-a.s., $v \in L^1([0, T]; W^{2,2}(M))$, then we say $v$ is a strong solution.

**Remark B.2.** *If $v$ is a strong solution, then it follows that*

$$v(t) = v_0 + \int_0^t \Delta_M v(s) ds + \mathcal{W}^Q t,$$

*where the equality is understood in $L^2(M)$.*

**Theorem B.3.** *There exists a unique strong solution of (1) with the following properties:*

*(i) The solution admits the representation*

$$v(t) = \sum_{k=0}^{\infty} \left( e^{-\lambda_k t} \langle v_0, \phi_k \rangle_{L^2(M)} + \frac{\sqrt{q_k}}{\lambda_k}(1 - e^{-\lambda_k t}) z_k \right) \phi_k \,, \tag{18}$$

*where $\{z_k\}_{k=0}^{\infty} \overset{\text{i.i.d.}}{\sim} \mathcal{N}(0,1)$;*

*(ii) If $\sum_{k=0}^{\infty} \lambda_k^{s-2} q_k < \infty$ for some $s \geq 2$, then for each $t > 0$, there is a positive constant $C_t$ that tends to infinity as $t \downarrow 0$ such that*

$$\mathbb{E}[|v(t)|_{W^{s,2}}^2] \leq C_t(\mathbb{E}[|v_0|_{L^2(M)}^2] + 1) \,;$$

*(iii) Conditional on $v_0$, the solution $v(t)$ is an $L^2(M)$-valued Gaussian random field with:*

- *Conditional mean: $\mathbb{E}[v(t) \mid v_0] = \sum_{k=0}^{\infty} e^{-\lambda_k t} \langle v_0, \phi_k \rangle_{L^2(M)} \phi_k$*

- *Conditional covariance operator: $K(t, s \mid v_0) : L^2(M) \to L^2(M)$ given by*

$$K(t, s \mid v_0) = \sum_{k=0}^{\infty} \frac{q_k}{\lambda_k^2}(1 - e^{-\lambda_k t})(1 - e^{-\lambda_k s})(\phi_k \otimes \phi_k) \,.$$

*Proof of Theorem 3.1 and B.3.* The existence and uniqueness of a weak solution for initial data satisfying $\mathbb{P}$-almost surely $v_0 \in L^2(M)$ follows from classical theory, combining Ball (1977) with Grigor'yan (2009)[Thm. 4.9]. We remark that additional comprehensive treatments for more general equations with space-time white noise can be found in Da Prato & Zabczyk (2014)[Thm. 5.4], Prévôt & Röckner (2007)[Thm. 4.2.4], and Elliott et al. (2012)[Prop. 2.6].

The space-time maximal regularity theory and continuity with respect to initial conditions follow from standard parabolic theory Sinestrari (1985); Amann (2016) with summaries in Da Prato & Zabczyk (2014)[App. A]). When $\mathbb{P}$-almost surely $v_0 \in W^{2,1}(M)$, following [9, Ch. 7 Thm. 5], we obtain $\mathbb{P}$-almost surely $v \in L^2([0,T]; W^{2,2}(M))$, confirming that $v$ is a strong solution.

By Ball (1977), the solution $v$ is also a mild solution satisfying $\mathbb{P}$-a.s. for all $t \in [0, T]$:

$$v(t) = \exp(\Delta_M t)v_0 + \int_0^t \exp(\Delta_M(t - s))\mathrm{d}s \mathcal{W}^Q \,.$$

Following Prévôt & Röckner (2007)[Prop. 2.1.6], the Gaussian random field $\mathcal{W}^Q$ admits the decomposition

$$\mathcal{W}^Q = \sum_{k=0}^{\infty} \sqrt{q_k} \phi_k z_k \,,$$

which converges in $L^2(\Omega; L^2(M))$. Substituting this into the mild solution formulation and using the eigenfunction expansion of the heat semigroup yields the representation (18), which proves (i).

Expectation solution estimates can be derived from analogous solution estimates in the deterministic theory. In particular, following Evans (2022)[Ch. 7 Thm. 5], one can obtain the variational bound:

$$\mathbb{E}\left[\sup_{t \leq T} |v(t)|_{W^{1,2}(M)}^2\right] + \mathbb{E}\int_0^T |v(t)|_{W^{2,2}}^2 dt \leq C \left( \mathbb{E}[|v_0|_{W^{1,2}(M)}^2] + \mathbb{E}[|\mathcal{W}^Q|_{L^2(M)}^2] \right) \,,$$

where $\mathbb{E}[|\mathcal{W}^Q|_{L^2(M)}^2] = \sum_{k=0}^{\infty} q_k$, and hence $v \in L^2(\Omega; L^{\infty}([0,T]; W^{1,2})) \cap L^2(\Omega; W^{2,2}(M))$.

To establish the $W^{s,2}$ estimates for $s \geq 2$, we employ a truncation argument. Define the finite-dimensional approximation:

$$v_K(t) := \sum_{k=0}^{K} \left( e^{-\lambda_k t} \langle v_0, \phi_k \rangle_{L^2(M)} + \frac{\sqrt{q_k}}{\lambda_k}(1 - e^{-\lambda_k t}) z_k \right) \phi_k \,.$$

Let $\mathfrak{s} \geq 2$. Using the Bessel-potential characterisation of Sobolev spaces on $M$ Behzadan & Holst (2022) and the fact that $\mathbb{E}[|z_k|^2] = 1$:

$$\mathbb{E}\left[|v_K(t)|^2_{W^{\mathfrak{s},2}}\right] = \mathbb{E}\left[\sum_{k=0}^{K}(1+\lambda_k)^{\mathfrak{s}}(v_K(t),\phi_k)^2_{L^2(M)}\right]$$

$$\leq \underbrace{2\,\mathbb{E}\left[\sum_{k=0}^{K}(1+\lambda_k)^{\mathfrak{s}}e^{-2\lambda_k t}(v_0,\phi_k)^2\right]}_{I_K} + \underbrace{2\sum_{k=0}^{K}(1+\lambda_k)^{\mathfrak{s}}\frac{q_k}{\lambda_k^2}(1-e^{-\lambda_k t})^2}_{II_K}.$$

By the power mean inequality, for all $\mathfrak{s} \geq 1$, $(1+\lambda_k)^{\mathfrak{s}} \leq 2^{\mathfrak{s}-1}(1+\lambda_k^{\mathfrak{s}})$. Since $\lambda_k \geq 0$, we have $(1-e^{-\lambda_k t})^2 \leq 1$ for all $t \geq 0$. Thus, since $Q$ is trace class, and importantly, $\sum_{k=0}^{\infty}\lambda_k^{\mathfrak{s}-2}q_k < \infty$, we can bound $II_K$ as:

$$II_K = \sum_{k=1}^{K}(1+\lambda_k)^{\mathfrak{s}}\frac{q_k}{\lambda_k^2}(1-e^{-\lambda_k t})^2 \leq \frac{2^{\mathfrak{s}-1}}{\lambda_1^2}\sum_{k=1}^{\infty}q_k + 2^{\mathfrak{s}-1}\sum_{k=1}^{\infty}\lambda_k^{\mathfrak{s}-2}q_k =: C_{II} < \infty,$$

which establishes the uniform boundedness of $II_K$ with respect to $K$.

To bound $I_K$, we define $f_{t,\mathfrak{s}} : \mathbb{R}_+ \to \mathbb{R}_+$ by $f_{t,\mathfrak{s}}(\lambda) = \lambda^{\mathfrak{s}}e^{-2\lambda t}$ and compute its derivative:

$$f'_{t,\mathfrak{s}}(\lambda) = \mathfrak{s}\lambda^{\mathfrak{s}-1}e^{-2\lambda t} - 2t\lambda^{\mathfrak{s}}e^{-2\lambda t} = \lambda^{\mathfrak{s}-1}e^{-2\lambda t}(\mathfrak{s}-2t\lambda).$$

This shows that $f_{t,\mathfrak{s}}$ attains its maximum at $\lambda = \frac{\mathfrak{s}}{2t}$ with maximum value $f_{t,\mathfrak{s}}\left(\frac{\mathfrak{s}}{2t}\right) = \left(\frac{\mathfrak{s}}{2te}\right)^{\mathfrak{s}}$.

Therefore,

$$\lambda_k^{\mathfrak{s}}e^{-2\lambda_k t} \leq \tilde{C}_t := \max\left(\max_{l:\lambda_l < \frac{\mathfrak{s}}{2t}}\lambda_l^{\mathfrak{s}}e^{-2\lambda_l t}, \left(\frac{\mathfrak{s}}{2te}\right)^{\mathfrak{s}}\right), \quad \forall t > 0, \quad k \in \mathbb{N}_0,$$

where we note that since $\lambda_k \to \infty$ as $k \to \infty$, there are only finitely many $k'$ such that $\lambda_{k'} < \frac{\mathfrak{s}}{2t}$. Indeed, by Weyl's law Chavel (1984)[Ch. 1, Eq. 50], there exists a univeral constant $C > 0$ independent of $M$ such that asymptotically $\lambda_k \sim \frac{C}{\nu(M)}k$, where $\nu(M)$ is the volume of $M$. Thus, for each $K \in \mathbb{N}$, we can bound $I_K$ as:

$$I_K = \sum_{k=0}^{K}(1+\lambda_k)^{\mathfrak{s}}e^{-2\lambda_k t}(v_0,\phi_k)^2 \leq 2^{\mathfrak{s}-1}(1+\tilde{C}_t)\sum_{k=0}^{K}(v_0,\phi_k)^2 \leq 2^{\mathfrak{s}-1}(1+\tilde{C}_t)|v_0|^2_{L^2(M)}.$$

Therefore, for each $K \in \mathbb{N}$, defining

$$C_t = 2\max(2^{\mathfrak{s}-1}(1+\tilde{C}_t), C_{II}),$$

we have the uniform bound:

$$\mathbb{E}\left[|v_K(t)|^2_{W^{\mathfrak{s},2}}\right] \leq C_t(\mathbb{E}[|v_0|^2_{L^2(M)}]+1).$$

Passing to the limit as $K \to \infty$ and using the weak lower-semicontinuity of the norm, we obtain

$$\mathbb{E}\left[|v(t)|^2_{W^{\mathfrak{s},2}}\right] \leq \liminf_{K\to\infty}\mathbb{E}\left[|v_K(t)|^2_{W^{\mathfrak{s},2}}\right] \leq C_t(\mathbb{E}[|v_0|^2_{L^2(M)}]+1).$$

The linearity of the stochastic heat equation ensures that conditional on the initial data $v_0$ (which, itself, may be random and non-Gaussian), the solution $v$ is a Gaussian process on $[0,T]\times M$. The conditional mean and covariance operator stated in part of the theorem follow directly from the spectral representation and the independence of the $\{z_k\}$ from $v_0$. This completes the proof of (iii), and the theorem. $\qquad\square$

**Remark B.4.** *It is worth highlighting the recent seminal work on Gaussian processes on manifolds Borovitskiy et al. (2020); Hutchinson et al. (2021); Rosa et al. (2023). While these papers focus on interpolating functions or tensor fields on a fixed Riemannian manifold, our work specifically needs to address the more challenging problem of inference across different manifolds. Probabilistic DiffusionNet is constructed by composing GP stochastic diffusion layers with MLPs, enabling transfer learning between shapes. An interesting direction for future work would be to explore cross-manifold inference purely from a GP perspective.*

## C  ENCODER FOR AMORTISED INFERENCE

When employing amortised variational inference for fitting PDN, we approximate the intractable posterior distribution of the latent variables $\{z_i\}_{i=1}^N$ given observations of the solution fields $\{y_{s_i}\}_{i=1}^N$, inputs $\{v_i\}_{i=1}^N$ and meshes $\{M_{s_i}^h\}_{i=1}^N$, with a variational distribution $q_\varphi$ defined in (15). The variational distribution is designed to be tractable and easy to sample to compute the ELBO (14).

There are a number of choices for $(\mu^\varphi, \Sigma^\varphi)$, and we provide an ablation study in Table 4. In particular, we consider the choices, where we drop the notation on the shape $s_i$ for convenience:

1. Partial Diffusion (PD), in which for each mesh $i \in \{1, \ldots, N\}$, channel $c \in \{1, \ldots, d_c\}$, and layer $l \in \{1, \ldots, L\}$,

$$\mu_l^\varphi := \mu_l^\varphi(y, M^h) = [\exp(-\Lambda \tau_c)\Phi^\top A \mathcal{L}_{\text{lift}}^\mu(y)]_{c=1}^{d_c}$$

$$\text{diag}(\Sigma_l^\varphi)(y, M^h)(y, M^h) = \left(\text{softplus}\left[[\exp(-\Lambda \tau_c)\Phi^\top A \mathcal{L}_{\text{lift}}^\Sigma(y)]_{c=1}^{d_c}\right]\right)^2$$

where $\Phi$, $\Lambda$, and $A$ denote the eigenvectors, eigenvalues, and mass matrix associated with the cotan-laplacian of $M^h$, as defined in Section 2.2. $\mathcal{L}_{\varphi_{\text{lift}}}^\mu, \mathcal{L}_{\varphi_{\text{lift}}}^\Sigma : \mathbb{R}^{d_{\text{in}}+d_{\text{out}}} \to \mathbb{R}^{d_c}$ are pointwise lifting layers with learnable parameters $\varphi_{\text{lift}}$, and $\tau_c \in \mathbb{R}_+$ is the learnable diffusion time associated with channel $c$. We use $\tau$ to distinguish the diffusion time parameter in the encoder from those of the base Diffusion-Net model. We call this *partial* diffusion block in the sense that $y$ is projected into the truncated eigenspace of the Laplace matrix $L^h$, before the diffusion multiplier and a learnable linear transformation are applied. Notice that (PD) suffers from basis ambiguities in that, for a given eigenvalue, the eigenvectors in a given eigenspace are defined only up to an orthogonal transformation, which reduces to sign ambiguities for simple eigenvalues (c.f. Lim et al. (2022) and Section A.1). We discuss this issue further below.

2. Diffusion followed by an aggregation layer (DN+), in which

$$\mu_l^\varphi(y, M^h) = [\mathcal{E}_{\phi_l}^\mu \circ \mathcal{P}_l \circ \mathcal{L}_{\varphi_{\text{lift}}}^\mu](y),$$

$$\text{diag}(\Sigma_l^\varphi(y, M^h) = [\mathcal{E}_{\phi_l}^\Sigma \circ \mathcal{P}_l^\tau \circ \mathcal{L}_{\varphi_{\text{lift}}}^\Sigma](y),$$

where we use $\mathcal{P}^\tau + l$ to denote the diffusion operation ((2)) for layer $l \in \{1, \ldots, L\}$ using diffusion times $\tau$, once again to distinguish between the diffusion times in the encoder and those of the DN blocks. $\mathcal{E}_{\phi_l}^\mu \in C(\mathbb{R}^{K \times d_c}; \mathbb{R}^{K \times d_c})$, $\mathcal{E}_{\phi_l}^\Sigma \in C(\mathbb{R}^{K \times d_c} \mathbb{S}_+^{K \times d_c})$ are aggregation layers. In practice, we use Perceiver cross-attention modules (Jaegle et al., 2021) to aggregate (see, also, Calvello et al. (2024)).

**Effect of PD's Basis Ambiguity:** Under eigenvector sign ambiguity, where $\phi_k \to -\phi_k$ for some modes, the encoder parameters transform as $[\mu_z]_k \to -[\mu_z]_k$ while the standard deviation becomes $\tilde{\sigma}[k] = \text{softplus}(-x_k)$ where $x_k$ denotes the pre-activation value. Using the identity $\text{softplus}(-x) = \text{softplus}(x) - x$, the standard deviation shifts additively. The variance, being the square of the standard deviation, transforms as:

$$\text{diag}(\tilde{\Sigma}_l^\varphi)[k] = (\text{softplus}(x_k) - x_k)^2 \tag{19}$$

Consequently, samples from the encoder under the flipped basis follow $\tilde{z}_k \sim \mathcal{N}(-[\mu_z]_k, \text{diag}(\tilde{\Sigma}_l^\varphi)[k]$. The stochastic diffusion operator incorporates the encoded latent variables as:

$$[\mathfrak{N}_{t,\eta,z}]_{ic} = \sum_{k=1}^K \lambda_k^{\eta_c-1}(1 - e^{-\lambda_k t_c})z_k[\Phi_s]_{ik} \tag{20}$$

When both encoder and decoder operate on the same eigenbasis within a forward pass, the sign ambiguity creates a compatible equivariance. Under the flipped basis, the noise term becomes:

$$[\tilde{\mathfrak{N}}_{t,\eta,\tilde{z}}]_{ic} = \sum_{k=1}^K \lambda_k^{\eta_c-1}(1 - e^{-\lambda_k t_c})\tilde{z}_k \tilde{\phi}_{k,i} \tag{21}$$

Table 4: Comparison of PD and DN+. We averaged the metric across all samples in the **test split** of the ShapeNet car dataset.

| Method | Par (M) | ↓RMSE | ↓NLL | ↓MCAL | ↓IS |
|--------|---------|-------|------|-------|-----|
| PD | 1.8 | **3.87** | **2.74** | 0.09 | 8.02 |
| DN+ | 7.3 | 4.03 | 3.21 | **0.08** | **7.87** |

Since $\tilde{z}_k$ has mean $-[\mu_z]_k$ and $\tilde{\phi}_k = -\phi_k$, the expected reconstruction satisfies:

$$\mathbb{E}[\tilde{z}_k \cdot \tilde{\phi}_k] = (-[\mu_z]_k) \cdot (-\phi_k) = [\mu_z]_k \cdot \phi_k \tag{22}$$

This demonstrates that the mean reconstruction in physical space remains invariant to eigenvector sign flips, as the sign changes in the latent encoding and eigenbasis cancel exactly.

While the mean reconstruction is preserved, the standard deviation of the stochastic component changes under basis ambiguity. The softplus parameterisation provides crucial stability properties. For small pre-activations $|x_k| \ll 1$, which are encouraged by KL regularisation, the softplus function exhibits near-symmetry: $\text{softplus}(x) \approx \log(2) + x/2$ and $\text{softplus}(-x) \approx \log(2) - x/2$. This yields standard deviations that differ by approximately $x$, creating bounded perturbations to the noise magnitude.

The variance of the SP operator's stochastic component scales as:

$$\text{Var}[\mathfrak{N}_{t,\eta,z}] \propto \lambda_k^{2(\eta_c-1)}(1 - e^{-\lambda_k t_c})^2 \cdot \text{Var}[z_k] \tag{23}$$

Under sign flips, while $\text{Var}[z_k]$ changes from $\text{diag}(\tilde{\Sigma}_l^\varphi)[k]$ to $(\text{softplus}(x_k) - x_k)^2$, this variation remains bounded and continuous, preventing the extreme fluctuations that would destabilise training. The KL divergence term in the ELBO further regularises this behaviour by penalizing large deviations from unit variance, naturally driving the system toward regions where $\text{softplus}(x) \approx \text{softplus}(-x)$.

The PD encoder thus achieves robust performance despite basis ambiguity through the interaction between the encoder's sign equivariance and the SP operator's multiplicative structure. The softplus nonlinearity ensures that variance perturbations under sign flips remain bounded, while the consistent mean reconstruction preserves the fidelity of the physical solution. This mechanism explains the empirical success of PD encoders, as the basis ambiguity effectively acts as a form of stochastic regularisation during training, with the model learning to operate in regimes where sign variations have minimal impact on the overall reconstruction quality.

(DN+) is not subject to such ambiguities. There are, of course, other possible variations in the encoder, such as choosing between data-amortisation, in which $\mu^\varphi, \Sigma^\varphi$ do not use geometric features $b_i$, geometry-amortisation, in which $\mu^\varphi, \Sigma^\varphi$ do not use the output field $y_i$, or no amortisation in which $\mu^\varphi, \Sigma^\varphi$ have no dependency on $s_i$. In Table 4, we apply data-amortisation, however we will explore variations in future work.

**Ablation Study**    In Table 4, we present an ablation of the choices (a) and (b). We compare their performance with the same accuracy and UQ metrics. In both cases, we take 16 PDN layers. In the case of (PD) we repeat the latent across all channels, whereas in (DN+) we learn mean and variance parameters for each channel. Whilst both methods yield similar performance, (PD) has $\sim 4\times$ fewer parameters and so we report its results in the main text as PDN. We anticipate that for larger datasets (DN+) will be more expressive than (PD). In future work we will explore this prospect, as well as considerations for reducing the parameter footprint of (DN+).

## D    IMPLEMENTATION DETAILS

**Preprocessing for (P)DN**    Following Li et al. (2023d), we used a standard Gaussian normalisation for the pressure fields. We also put all the meshes within a $[-1, 1]$ bounding box, again following Li et al. (2023d).

**Optimisation for (P)DN**   We used the AdamW (Loshchilov & Hutter, 2019) with learning rate (LR) set to $10^{-3}$ and weight decay set to $10^{-4}$. We used the One Cycle LR scheduler with the same hyperparameters as was used in Wu et al. (2024). We trained each model for 200 epochs. For PDN we evaluated the Monte Carlo estimate of the gradient of the ELBO using just a single sample from the encoder.

**Architectural specifications of (P)DN**   For (P)DN we used $K = 128$ modes, and $L = 16$ layers throughout. We chose the number of channels to be the same size as the number of modes ($K = d_c = 128$), following Sharp et al. (2022).

**Preprocessing/Optimisation/architectural setup for other baselines**   We made sure for every baseline we follow the same pressure field normalisation as described above. For optimisation and other geometric preprocessing we retained exactly the same settings as was originally proposed for that particular baseline in its original publication or released code. We did the same for the architecture. Specifically, while using Transolver as a base model, for the UQ experiments, we retained the exact same architecture as was used for the results in Table 3 of Wu et al. (2024). For GINO we used the same architecture as the "decoder" version in Table 2 & 3 in Li et al. (2023d), respectively.

## E    EXPLANATION OF THE METRICS

To measure predictive accuracy, we use the the relative $L^2$ norm which, for a given observation $y \in \mathbb{R}^{n_s \times d_{\text{out}}}$ and prediction $\widehat{y} \in \mathbb{R}^{n_s \times d_{\text{out}}}$ on a mesh $M^h$, is given by

$$\mathrm{RL2}(y, \hat{y}) = \frac{\|y - \widehat{y}\|_2}{\|y\|_2}.$$

This is a metric is widely used in the neural operator literature, allowing us to compare (P)DN with other methods. However, for comparing UQ performance it is important to assess methods in a holistic way. Thus, we have used a number of metrics which we describe next.

We use the following notation: Given a test set mesh $i \in \{M_1^h, \ldots, M_{N_{\text{test}}}^h\}$ and vertices $X_i = [x_{i1}, \ldots, x_{in_i}]^\top \in \mathbb{R}^{n_i \times 3}$ where $n_i = |\mathcal{V}_i^h|$ is the number of vertices in $M_i^h$, let $y_{ij}$ denote the evaluations of the observed output function at point $x_{ij} \in V_i^h$. For baseline models that support uncertainty quantification via sampling, $\widehat{\mu}_{ij} \in \mathbb{R}^{d_{\text{out}}}$ represents the empirical mean over samples, and $\widehat{\sigma}_{ij} \in \mathbb{R}_+^{d_{\text{out}}}$ denotes the empirical standard deviation. For convenience, we collect $y_i = \{y_{ij}\}_{j=1}^{n_i}$, $\widehat{\mu}_i = \{\mu_{ij}\}_{j=1}^{n_i}, \widehat{\sigma}_i = \{\sigma_{ij}\}_{j=1}^{n_i}$.

Note that the empirical mean, standard deviations and other credible intervals were all obtained using 100 samples from the predictive distributions of respective models.

**Negative log likelihood (NLL)**   We used the Gaussian negative log likelihood of the predicted field given by

$$\mathrm{NLL}(y_i, \widehat{\mu}_i, \widehat{\sigma}_i) = -\frac{1}{n_i} \sum_{j=1}^{n_i} \ln \mathcal{N}(y_{ij} | \widehat{\mu}_{ij}, \widehat{\sigma}_{ij}^2 I_{d_{\text{out}}}).$$

**Total Miscalibration Area**   The Miscalibration Area (MCAL) measures the discrepancy between predicted confidence levels and the observed frequencies of events. In particular, let $\pi^{(p)}$ denote expected coverage levels for $p \in [0, 1]$. For each input $i \in [N_{\text{test}}]$ and channel $c \in [d_c]$, the observed coverage at level $\pi^{(p)}$ is:

$$\hat{\pi}_{i,c}^{(p)} = \frac{1}{n_i} \sum_j^{n_i} \mathbb{I}\left( \frac{|y_{ij,c} - \hat{\mu}_{ij,c}|}{\widehat{\sigma}_{ij,c}} \leq \Phi^{-1}\left( \frac{1 + \pi^{(p)}}{2} \right) \right)$$

where $\Phi$ is the cumulative distribution function of the standard normal and $\mathbb{I}(\cdot)$ is the indicator function. The miscalibration area is given by

$$\mathrm{MCAL}(y_i, \widehat{\mu}_i, \widehat{\sigma}_i) = \frac{1}{d_c} \sum_{c=1}^{d_c} \int_0^1 |\widehat{\pi}_{i,c}^{(p)} - \pi^{(p)}| \, dp.$$

In practice, the integral may be approximated through the trapezoidal rule Chung et al. (2021).

**Interval Score**  Interval score, proposed in Gneiting & Raftery (2007), is a *proper scoring rule* (see definition in Chung et al. (2021)) that is a summary statistic of overall performance of a distributional prediction. In particular the formula for IS is given by

$$\text{IS}(y_i, L_i, U_i) = \frac{1}{n_i \cdot d_{\text{out}}} \sum_{j,c=1}^{n_i, d_{\text{out}}} \Big[ (U_{ij,c} - L_{ij,c}) + 2 * \alpha * (L_{ij,c} - y_{ij,c}) * \mathbb{I}(y_{ij,c} < L_{ij,c})$$

$$+ 2 * \alpha * (y_{ij,c} - U_{ij,c}) * \mathbb{I}(y_{ij,c} > U_{ij,c}) \Big],$$

where $\alpha$ is a parameter representing the level of the prediction interval; $U_i, L_i$ are the upper and lower bounds of the prediction interval, and $\mathbb{I}(\cdot)$ is the indicator function. We calculated $U_i, L_i$ by drawing 100 samples of $\widehat{y}_i$, using a particular chosen $\alpha$. We chose $\alpha = 0.025$ throughout.

In all probabilistic experiments, we generated 100 samples per prediction, and the reported results are averaged over all $N_{\text{test}}$ shapes in the test set.

**Why multiple UQ metrics?**  When extracting probabilistic predictions for UQ one ideally wants good coverage (smaller calibration error) so that we can associate regions (in the input space, for us the mesh vertices) of high uncertainty with erroneous predictions. However, poor predictive performance can also have good calibration since an underperforming model can have good coverage with large credible intervals. Thus it is necessary to also access *sharpness* which quantifies the concentration of the predictive distribution. The best models should be competitive in terms of both sharpness and calibration. Thus, given the inherent trade-off between sharpness and calibration we chose multiple metrics. We assessed calibration through the total miscalibrated area, coverage and sharpness through the interval score.

# F  DISCUSSION ON THE CHOICE AND IMPLEMENTATION OF CLASSICAL UQ METHODS

**Choice of the base models**  Transolver (TS) and GINO have emerged as state-of-the-art approaches towards building CFD surrogates for industrial scale problems, based on recent benchmarking studies Li et al. (2023d); Pepe et al. (2025); Wu et al. (2024). Moreover, they propose contrasting, but foundational, approaches (e.g. attention mechanism for Transolver) towards building an integral transform layer, for achieving resolution invariance, that can serve as building blocks for similar architectures. UQ performance with these base models, thus would be indicative of UQ capabilities of similar base models. Finally, we have also considered DN as a base model to highlight the benefits of PDN.

**Choice of UQ methods**  We have specifically chosen those UQ methods that have recently been used in the context of neural operators. Laplace Approximation have been applied recently to endow a base FNO model, operating on a regular domain, with UQ capability Weber et al. (2024); Magnani et al. (2022). Model ensembling for extracting uncertainty (Lakshminarayanan et al., 2017), in the context of FNOs on regular domain, has been explored recently in Li et al. (2024) towards building an active learning framework, and was shown to outperform other popular approaches for UQ. In addition to these, we have also considered Monte Carlo Dropout, which is arguably the most popular approach for carrying out UQ for any deep neural network.

We like to point out that we indeed try to implement the Conformal Prediction (CP) approach for NOs proposed in Ma et al. (2024b). However, we could not reproduce the results in Ma et al. (2024b). Due to the lack of access to the officially released code, we decided against includ CP in our benchmarking. However, note that: i) our results already show very good calibration out-of-the-box, at least for ShapeNet cars, and ii) CP can be easily added on top of our method as an additional calibration technique (see Fong & Holmes (2021)).

**Laplace Approximation**  For the Laplace Approximation, we used a last-layer approximation. In particular, we fit the Hessian of a Negative Log-Likelihood loss, using a learned homoscedastic noise

variance $\sigma^2$, with respect to the parameters $\theta_f$ of the final (linear) layer of each baseline model

$$\mathcal{L}(\theta_f) = -\sum_{i=1}^{N} \ln \mathcal{N}(y_i | \mu_{\theta_f}, \sigma^2),$$

where $y_i$ is the observed output field i.e. the pressure field on the mesh vertices, and $\mu_{\theta_f}$ is the corresponding model (DN, Transolver, GINO) prediction, and $m$ is the total number of shapes in the training dataset.

We approximate the Hessian with the Fisher Information matrix, which is the outer product of the gradients. Following Ritter et al. (2018), we scale the Hessian and add a multiple of the identity to yield

$$H_{M,\tau} = M\mathbb{E}\left[\frac{\partial^2 \log \mathcal{L}(\theta_f)}{\partial \theta_f^2}\right] + \tau I$$

where the expectation is approximated through Monte-Carlo samples and $M, \tau$ are hyperparameters to tune on a validation set. In our case we choose $N = 10^8$, $\tau = 500$.

**Monte Carlo Dropout** Monte Carlo Dropout (DO) is one of the most widely used methods to carry out UQ for a deep neural network. We apply dropout both at the time of training and inference. In Gal & Ghahramani (2016), it was shown that training a neural network while applying dropout is equivalent to variational inference of the posterior distribution of the network weights. At test time, a forward pass through the model, with Dropout, is equivalent to a sample from the posterior predictive distribution of the network output, for us the pressure field. The role of a prior on weights is fulfilled by applying a $L^2$ regularisation: $p(\theta) \propto w_d \|\theta\|^2$, where $w_d$ is a hyperparameter. In practise, the prior is introduced through controlling the weight-decay hyperparameter in AdamW (Loshchilov & Hutter, 2019).

For all the baselines, we train using the same Gaussian Negative Log-Likelihood loss, using a learned homoscedastic noise variance $\sigma^2$, a dropout probability of $p = 0.1$ and as weight-decay of $w_d = 10^{-4}$.

For Transolver DO was applied within the attention and projection layers, for GINO DO was applied on MLPs within the FNO blocks, and for DN this was applied to the MLPs within each diffusion block.

**Model Ensemble** Model Ensemble (ME) is a computationally intensive, yet conceptually simple approach for carrying out UQ. ME constitutes ensembling the prediction of the same base model, however trained with different parameter initialisations. The intuition behind this approach is that by exploring a basin of attraction, in the parameter space, through multiple local minima, one can obtain a region in the parameter space each of whose points will generate a model output that would correspondingly live within a "confidence" band, thus quantifying uncertainty.

In practise, as suggested in Lakshminarayanan et al. (2017), we trained all baselines using a a Gaussian Negative Log-Likelihood loss, with a heteroskedastic noise i.e. with a spatially-varying variance $\sigma_\theta^2 = \sigma_\theta^2(x)$, where $x$ are the vertices of a mesh and $\theta$ are the model parameters. To generate the spatially-varying mean and variance, we used two heads, as was proposed in Li et al. (2024), for the last layer of all baselines. The resulting loss is given by

$$\mathcal{L}(\theta) = -\ln \mathcal{N}(y | \mu_\theta, \sigma_\theta^2).$$

The predictive distribution is then given by the following mixture distribution

$$\frac{1}{P}\sum_{p=1}^{P} \mathcal{N}(\mu_{\theta^{(p)}}, \sigma_{\theta^{(p)}}^2),$$

where $\theta^{(p)}$ is the learned parameter corresponding to the $p$-th model initialisation. We set $P = 10$ for all experiments.

## G    ADDITIONAL RESULTS

In total we performed three repeats of all the UQ experiments barring the ones including model ensemble (since this method already includes multiple repeats of optimization with the same model).

In Tables 5- 7, we report the UQ metrics summarized as mean across the test set samples. In Table 2, we furnish the average of these metrics, across all three repeats, for the methods that were repeated. Note the average ranks were calculated using the metrics furnished in Table 2 only.

In addition to the baselines mentioned in the main text, we have also considered a recently proposed sparse attention based architecture Erwin (Zhdanov et al., 2025). In particular, we used Erwin with $\approx 10^6$ parameters (Erwin small, EWS) and with $\approx 5 \times 10^6$ parameters (Erwin medium, EWM). We paired these models with DO based UQ. We found Erwin's performance to be noticeably poor for the Ahmed dataset, and thus we decided to not include Erwin's results on Ahmed. Since, we have only included Erwin's results for Shapenet, thus we have decided to furnish Erwin's results in the appendix only. For Erwin we did a single repeat of the experiments.

Table 5: **First Repeat:** UQ metrics for surface pressure field prediction.

| Methods | ShapeNet car | | | | Ahmed bodies | | | |
|---|---|---|---|---|---|---|---|---|
| | RMSE↓ | NLL↓ | MCAL↓ | IS↓ | RMSE↓ | NLL↓ | MCAL↓ | IS↓ |
| GINO-DO | 6.61 | **2.46** | **0.08** | 10.05 | 33.51 | 5.26 | 0.25 | 127.96 |
| GINO-LA | 4.35 | 27.2 | 0.23 | 15.25 | 35.37 | 5.29 | 0.25 | 113.54 |
| TS-DO | 3.88 | _2.47_ | 0.11 | 9.64 | 21.11 | _4.62_ | 0.3 | 120.33 |
| TS-LA | 4.03 | 6.07 | _0.09_ | 11.28 | 25.01 | 5.29 | 0.26 | 100.66 |
| DN-DO | 5.25 | 2.96 | 0.22 | 24.75 | 24.62 | 4.93 | 0.33 | 197.45 |
| DN-LA | 3.93 | 6.96 | _0.09_ | 11.16 | _20.47_ | 4.71 | **0.23** | **65.17** |
| EWS-DO | _3.82_ | 2.54 | 0.11 | 8.74 | – | – | – | – |
| EWM-DO | **3.66** | 2.51 | 0.10 | _8.47_ | – | – | – | – |
| PDN (Ours) | 3.87 | 2.74 | _0.09_ | **8.02** | **18.34** | **4.21** | _0.24_ | _68.28_ |

Table 6: **Second Repeat:** UQ metrics for surface pressure field prediction.

| Methods | ShapeNet car | | | | Ahmed bodies | | | |
|---|---|---|---|---|---|---|---|---|
| | RMSE↓ | NLL↓ | MCAL↓ | IS↓ | RMSE↓ | NLL↓ | MCAL↓ | IS↓ |
| GINO-DO | 6.61 | **2.46** | **0.08** | 10.05 | 33.51 | 5.26 | 0.25 | 127.96 |
| GINO-LA | 4.17 | 32.21 | 0.23 | 14.30 | 33.48 | 5.26 | _0.24_ | 104.66 |
| TS-DO | **3.88** | _2.50_ | 0.11 | _9.7_ | 21.57 | _4.65_ | 0.3 | 120.67 |
| TS-LA | 3.99 | 5.39 | _0.09_ | 11.03 | 25.01 | 5.29 | 0.26 | 100.66 |
| DN-DO | 4.71 | 2.82 | 0.22 | 21.50 | 28.34 | 4.93 | 0.266 | 185.14 |
| DN-LA | 4.30 | 7.36 | _0.09_ | 12.72 | **20.28** | 4.67 | _0.24_ | **66.31** |
| PDN (Ours) | _3.93_ | 2.74 | _0.09_ | **8.20** | _18.45_ | **4.27** | **0.23** | _66.85_ |

Table 7: **Third Repeat:** UQ metrics for surface pressure field prediction.

| Methods | ShapeNet car | | | | Ahmed bodies | | | |
|---|---|---|---|---|---|---|---|---|
| | RMSE↓ | NLL↓ | MCAL↓ | IS↓ | RMSE↓ | NLL↓ | MCAL↓ | IS↓ |
| GINO-DO | 6.56 | **2.42** | **0.07** | 9.94 | 31.2 | 5.32 | **0.23** | 109.57 |
| GINO-LA | 4.29 | 28.49 | 0.23 | 14.90 | 34.89 | 5.27 | 0.25 | 112.73 |
| TS-DO | **3.83** | _2.46_ | 0.11 | _9.38_ | _20.83_ | _4.58_ | 0.3 | 117.66 |
| TS-LA | 4.03 | 6.07 | 0.1 | 11.27 | 25.01 | 5.29 | 0.26 | 100.66 |
| DN-DO | _3.88_ | 2.51 | 0.2 | 15.41 | 25.94 | 4.91 | 0.31 | 187.95 |
| DN-LA | 4.32 | 8.25 | 0.11 | 13.08 | 21.63 | 4.79 | _0.24_ | **69.55** |
| PDN (Ours) | 3.93 | 2.83 | _0.08_ | **7.77** | **18.05** | **4.21** | **0.23** | _66.86_ |

## G.1 RESULTS ON NON-WATERTIGHT SHAPENET DATASET SPLIT:

Although our proposed approach relies on the assumption of a compact manifold, we relax that assumption in practise as long as we can evaluate the Laplacian. In particular, we apply the method proposed in Sharp & Crane (2020) which can operate on non-manifold shapes. As a result, we were able to use the full ShapeNet car dataset (Umetani & Bickel, 2018) with 789 shapes for training and 100 shapes for test, using the same train-test split as was used in Wu et al. (2024). We report the results for PDN in Table 8. We like to point out that all models were trained using not just the surface pressure field, but also the velocity field in the domain, putting PDN at a disadvantage. Also note that we only report pressure drag and ingore the skin friction component. Nevertheless PDN shows

SOTA performance as it outperforms most baseline methods with a fraction of the parameters (c.f. Table 10).

Table 8: Performance comparison in terms of relative $L^2$ distance, between ground truth pressure field on the surface of the shape and its prediction, and relative $L^2$ error for drag coefficient $C_d$, using the full **ShapeNet car** dataset, used in Wu et al. (2024). Both training and test sets have non-manifold meshes. Except PDN all models were trained using both pressure and velocity information.

| Methods | ↓Relative $L^2$ for surface pressure | ↓Relative $L^2$ for $C_d$ |
|---|---|---|
| GraphSage (Hamilton et al., 2017) | 0.1050 | 0.0270 |
| PointNet (Charles et al., 2017a) | 0.1104 | 0.0298 |
| Graph U-Net (Gao & Ji, 2019) | 0.11024 | 0.0226 |
| MeshGraphNet Pfaff et al. (2020) | 0.0781 | 0.0168 |
| GNO (Li et al., 2020a) | 0.0815 | 0.0172 |
| GeoFNO Li et al. (2023b) | 0.2378 | 0.0664 |
| GINO (Li et al., 2023d) | 0.0810 | 0.0184 |
| Galerkin (Cao, 2021) | 0.0878 | 0.0179 |
| GNOT (Hao et al., 2023) | 0.0798 | 0.0178 |
| 3D-GeoCA (Deng et al., 2024) | 0.0779 | 0.0159 |
| Transolver (Wu et al., 2024) | **0.0745** | **0.0103** |
| PDN (Ours) | 0.0752 | 0.0168 |

### G.2  ABLATIONS ON MODEL SIZE

In Table 9, we report an ablation study on the number of eigenmodes $K$ and number of PDN layers $L$, evaluated on the ShapeNet car dataset. We find that there is a clear increase in performance across all UQ metrics as both hyperparameters increase. In the paper we report results for $L = 16$, $K = 128$.

Table 9: Ablations of the number of probabilistic diffusion blocks $L$ and number of eigenmodes $K$ in PDN. We averaged the metric across all samples in the **test set** of the ShapeNet car dataset.

| Hyperparameters | ↓RMSE | ↓NLL | ↓MCAL | ↓IS |
|---|---|---|---|---|
| $L = 4, K = 128$ | 4.24 | 2.82 | 0.17 | 10.9 |
| $L = 8, K = 128$ | 4.06 | 3.34 | **0.09** | 10.42 |
| $L = 16, K = 128$ | 3.87 | 2.74 | **0.09** | **8.02** |
| $L = 4, K = 256$ | 3.90 | 2.61 | 0.15 | 12.52 |
| $L = 8, K = 256$ | **3.76** | 2.59 | 0.10 | 8.95 |
| $L = 16, K = 256$ | **3.76** | **2.48** | 0.12 | 9.66 |

### G.3  COMPUTATIONAL ASPECTS

In Table 10 we report the number of parameters of each model, alongside the allocated memory (GB) and average seconds per epoch during training. We found that for smaller meshes (ShapeNet cars, $\sim 3500$ vertices) PDN, although being memory efficient, is slower than the baselines. However, as the size of meshes (Ahmed bodies, $\sim 100,000$ vertices) grow, PDN runs much faster than the baselines, making it ideal for larger meshes typically found in industrial CFD applications.

In Table 11 we furnish the total time incurred in calculating the operators and caching them, subsequently training and carrying out inference using PDN. Note that each prediction on a test geometry requires 100 forward passes of the model to generate random draws of the pressure field. It should be also be noted that the cotan-Laplacian of a closed 2-manifold is a PSD sparse matrix, and so is amenable to preconditioning methods to accelerate eigendecomposition. Indeed, we employ a Cholesky preconditioning which allows us to eigendecompose Laplacians for meshes of $10^6$ nodes in $< 6$ seconds on GPU, rendering the method suitable for typical industrial applications.

Table 10: Efficiency comparison in terms of parameters, average seconds per epoch and memory (GB) allocated during a training epoch.

| Methods | The **ShapeNet car** dataset | | | The **Ahmed bodies** dataset | | |
| | Parameters (M) | Time (s/epoch) | Memory (GB) | Parameters (M) | Time (s/epoch) | Memory (GB) |
| --- | --- | --- | --- | --- | --- | --- |
| GINO | 366 | **17.86** | 18.69 | 366 | 237.2 | **18.15** |
| TS | 3.8 | 18.91 | 1.29 | 3.8 | 403.12 | 52.1 |
| PDN (Ours) | **1.8** | 34.6 | **0.70** | **1.8** | **88.29** | 22.34 |

Table 11: Computation times for PDN, comprising of Operator caching (OP Caching), incurred training time and inference time, considering 100 draws of the pressure field from the predictive distribution.

| Methods | The **ShapeNet car** dataset | | | The **Ahmed bodies** dataset | | |
| | OP Caching (hours) | Training (hours) | Inference (hours) | OP Caching (hours) | Training (hours) | Inference (hours) |
| --- | --- | --- | --- | --- | --- | --- |
| PDN (Ours) | 0.18 | 1.92 | 0.21 | 1.77 | 5 | 0.5 |

### G.4 FURTHER ANALYSIS OF PDN'S OOD PREDICTIONS

To further probe PDN's out-of-distribution predictions we have carried out a comparison between PDN's predicted dispersion (which we define as sum of predictive variance over the surface) as obtained on 10 randomly drawn test shapes from an out-of-distribution versus the in-distribution splits of the Shapenet cars. We carry out the comparison using a scatter-plot (Fig. 3) between RMSE and dispersion values for each of the test shapes. The scatter-plot in clearly reveals that the uncertainty grows with out-of-distribution shapes.

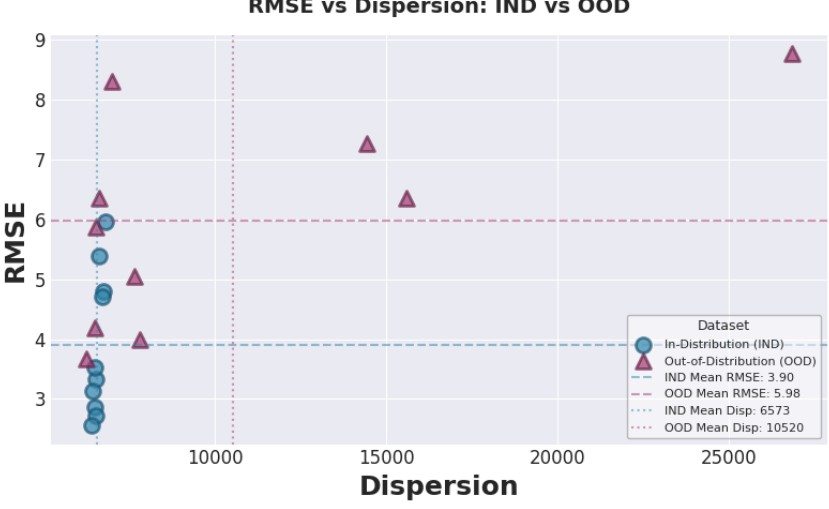

Figure 3: Comparison of PDN's performance (RMSE) versus prediction uncertainty (dispersion) for in-distribution (IND, blue circles) and out-of-distribution (OOD, red triangles) test sets. Dashed and dotted lines indicate mean RMSE and dispersion values, respectively. OOD samples exhibit both higher error rates and greater variation in dispersion, highlighting PDN's ability to increase uncertainty on unseen data distributions.

### G.5 CONVERGENCE ANALYSIS OF THE NOISE SCALES $\eta$

In Fig. 4 we have shown a plot of the histogram (across channels) of learned values of $\eta$. Note that we initialised training as $\eta_c = 1$, for each channel, across all blocks. The learned negative values of $\eta$, on average, across all channels, and per block, reveal that PDN has discovered a physically-motivated uncertainty structure. In the stochastic diffusion operator, the noise term $[\mathfrak{N}_{t,\eta,z}]_{kc} = \lambda_k^{\eta_c-1}(1 - e^{-\lambda_k t_c})z_{kc}$ is weighted by $\lambda_k^{\eta_c-1}$, which with negative $\eta_c$ creates inverse spectral weighting: low-frequency modes with small eigenvalues receive large noise amplitudes, while high-frequency modes with large eigenvalues are strongly suppressed. This learned pattern indicates the model expresses uncertainty primarily about large-scale, smooth features rather than fine details. Effectively, the stochastic diffusion operator, Eq. (7), has learned to incorporate a low-pass stochastic perturbation, with similar scale across the layers.

## H BROADER IMPACT

This study presents a probabilistic extension for a popular graph neural network architecture, while providing solid theoretical footing for its interpretation as a neural operator for approximating PDE solutions on domain boundaries. The probabilistic extension allows us to provide uncertainty estimates without relying on post hoc methods which might struggle in the operator learning setting. Through robust benchmarking, we show that our approach has significant advantages and, together with our theoretical contributions, our work has the potential to advance computational modelling in scientific and engineering disciplines.

Given the importance of quantifying uncertainty for downstream predictions used for decision-making, how we are able to achieve that with a more efficient model, and how we provide theoretical grounding for further improvements, we recognise that our method could be employed in a wide range of high-stake applications and in sensitive contexts.

As the field adopts our contributions and advances, it will be essential to ensure that methods are used responsibly, particularly in applications where predictive reliability and interpretability are critical. Ongoing dialogue among machine learning researchers, domain experts, and the general public is essential to ensure that the development and deployment of such tools remain aligned with societal values and serve the public interest.

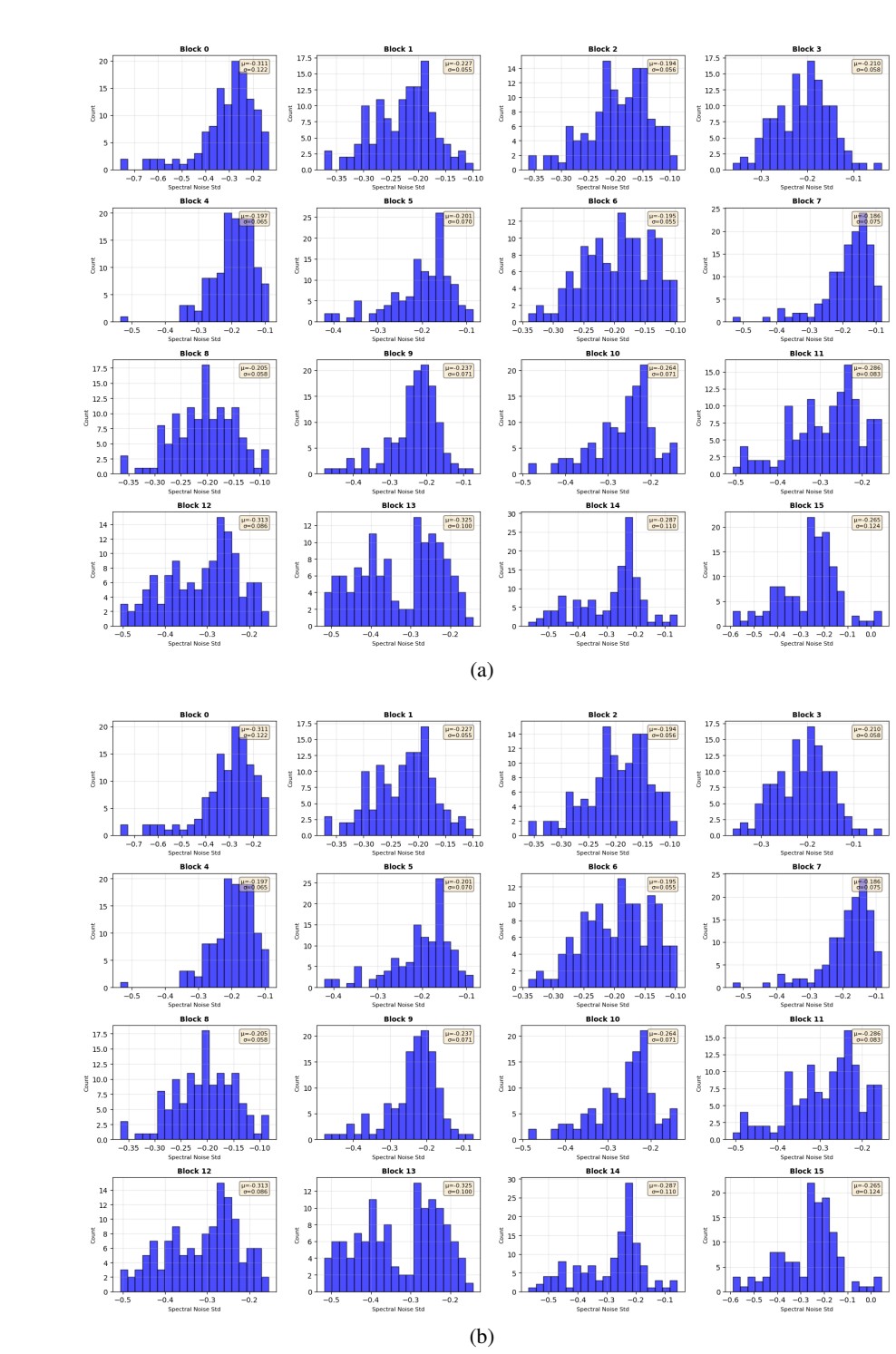

Figure 4: Distributions of learned noise scales $\eta$, across channels, for each block of PDN. (a) Shapenet car and (b) Ahmed datasets respectively.

