# OpenReview forum: "Probabilistic DiffusionNet: A geometry informed probabilistic generative surrogate for PDEs"
_ICLR.cc/2026/Conference — Submitted to ICLR 2026_

### Official Review · Reviewer_1PJd · 2025-10-22

**Soundness:** 4
**Presentation:** 4
**Contribution:** 4
**Rating:** 8
**Confidence:** 5

**Summary:**

This paper proposes a methodology for producing uncertainty quantifying surrogate models where geometry plays a key role in the field to be estimated. They base their methodology on pre-existing GNNs based on eigen-decompositions of the discretized Laplace-Beltrami operator. The aim is a surrogate model whose confidence should vary on unseen data. The method is test on CFD data on car geometries and shows to predictive accuracy with high quality UQ.

**Strengths:**

- The paper focuses on a very important, underexplored problem, in surrogate modelling for PDEs.
- The proposed method is very well explained with great clarity.
- Interesting and relevant theoretical results are reported.
- Care is taken in conceptually comparing and contrasting the proposed method to existing approaches, and the special cases where these overlap.
- The discussions in the appendix around UQ alternatives is usefull.

**Weaknesses:**

- As mentioned in the conclusion, eigen-decompositions of large cotangent-Laplacian matrices can get quite expensive and will grow like $O(n^3)$. In light of this, I would think it useful to report some compute cost metric, such as train time and prediction time for example (or some other suitable metric). The method might still be worth the cost if much more expensive, but this is important information which might help practitioners decide which tool is right for their problem. Can you add this evaluation to the manuscript? Some information is provided is Table 7, although this is limited and does not cover all experiments.

- Mentioned in the paper is the idea that UQ in this context is useful to help practitioners trust outputs from a model. It would be good have a test that shows high uncertainty for a collection of geometries which is considered "out of distribution".

- Studying another class of PDEs would be useful. For example, the Helmholtz problem, where the solution field depends very strongly and nonlocally on the geometry. Problems in elasticity and shell deformation would also be of interest to practitioners.

**Questions:**

- line 75: "Graph Neural Graph Neural Network"

- Can the relevance of the theoretical results be better explained and how they relate to practice.

- Did the diagonal approximation in the variational posterior cause any issues? Do you expect any gains to be made from better variational approximations?

---

> ### Author Response · Authors · 2025-12-03
> **Further timing details for PDN added, visualisation showing uncertainty increase for OOD geometry added**
>
> We thank the reviewer for carefully reading our manuscript. Our responses to the reviewer's questions/criticism are as follows:
>
> **eigen-decompositions of large cotangent-Laplacian matrices**
>
> We have now furnished (Table 11, Appendix G.3) the total time incurred in calculating the operators and caching them, subsequently training and carrying out inference using PDN. Also, see our response to a similar point made by reviewer nqZA.
>
> **It would be good have a test that shows high uncertainty for a collection of geometries which is considered "out of distribution"**
>
> We extended the results on OOD geometries (see §4.3) by including a comparison between PDN’s predicted dispersion (which we define as sum of predictive variance over the surface) as obtained on $10$ randomly drawn test shapes from an out-of-distribution versus the in-distribution splits of the Shapenet cars. We carry out the comparison using a scatter-plot (Figure 3) between rmse and dispersion values for each of the test shapes. The scatter-plot clearly reveals that the uncertainty grows with OOD shapes. This additional analysis is presented in Appendix G.4 in the revised manuscript.
>
> **Studying another class of PDEs would be useful**
>
> See our response (**paper's validation on a single class of physical problems...**) to a similar question raised by reviewer nqZA.
>
> **Can the relevance of the theoretical results be explained?**
>
> Theorem 3.1 is central to our reformulation of the DiffusionNet architecture.
> In particular, DiffusionNet takes advantage of the closed form solution to
> the deterministic heat equation in the basis of the Laplace-Beltrami Operator's (LBO)
> eigenfunctions. The learnable diffusion time parameter acts within this basis.
> To make our adaptation, we must similarly obtain the solution to the _stochastic_
> heat equation, also defined in the LBO eigenbasis. This is given in Eqn. (6), and
> motivates the resulting stochastic diffusion operator in Eqn. (7)
>
> In the appendix, we go on to determine an upper bound on the expected square norm of
> the output of a stochastic diffusion block, in terms of the expected square norm of
> the input. This guarantees stability of the stochastic diffusion operator that is
> central to the PDN block. Were this to not be the case, it may be possible that the
> solutions drawn from a PDN block are unbounded during training or inference.
>
> Finally, we also show that, on a given manifold, the solution of a PDN block defines
> a Gaussian random field, and we give an expression for the mean and covariance. This
> connects our work to recent works on Gaussian processes (GP) on manifolds and presents a
> direction for future analysis to explore the transfer learning capacity of DiffusionNet
> _across_ manifolds using a GP perspective.
>
>
> **Did the diagonal approximation in the variational posterior cause any issues?**
>
> The reviewer correctly points to possible limitations in the approximate inference scheme employed, and more specifically the choice for the approximating variational distribution. The empirical results suggest that the choice of uncorrelated times between blocks, but full diagonal correlation matrix between channels in a block, is sufficient for high-quality UQ. However, more may be gained by introducing a fuller correlation structure between blocks, or a more flexible approximating family, e.g. a mixture distribution. The need for this may depend on the problem at hand, but we expect the choice we made to serve as a robust default across settings, evidenced by the solid performance across our experiments.

---

### Official Review · Reviewer_ioK4 · 2025-10-28

**Soundness:** 3
**Presentation:** 3
**Contribution:** 3
**Rating:** 6
**Confidence:** 3

**Summary:**

The paper presents a method based on DiffusionNet for predicting distributions on surfaces. The authors modify DiffusionNet with a stochastic Diffusion Operator to allow for uncertainty quantification. The results show that PDN is competitive with current geometric baselines in both accuracy and UQ metrics.

**Strengths:**

- The perspective of discretization invariance from diffusion is interesting
- The results on uncertainty quantification are good, when compared to models that are not ensembled.
- The theoretical derivations are well explained and seem to be well motivated

**Weaknesses:**

- The primary weakness is that the model does not outperform current baselines, such as Transolver. Additionally, newer geometric learning baselines (ABT-UPT (https://arxiv.org/pdf/2502.09692), Erwin Transformer (https://arxiv.org/abs/2502.17019)) have been proposed which outperform prior Transolver/GINO/GNN-based methods on ShapeNetCar and more complex CFD datasets.
    - This isn’t necessarily a deal-breaker, since PDN/DN offers a new perspective on surrogate models, but is a weakness.
- The prediction seems to only be for surface values (surface pressure, wall shear stress), however, in practice, volumetric fields such as velocity/pressure are also important and are jointly predicted by other baselines

Overall, I think the idea is presented well and is interesting, although it does not outperform current methods.

**Questions:**

- There seems to be a typo on line 380, should ‘PDE’ be ‘PDN’
- The reference : " Anima Anandkumar, Kamyar Azizzadenesheli, Kaushik Bhattacharya, Nikola Kovachki, Zongyi Li,
Burigede Liu, and Andrew Stuart. Neural operator: Graph kernel network for partial differential
equations. In ICLR 2020 workshop on integration of deep neural models and differential equations,
2020" seems to have the wrong author order (https://arxiv.org/abs/2003.03485)
- I am curious about wall shear stress predictions and drag/lift prediction (from integrating x or y component of WSS and pressure forces along the surface).
    - Drag is commonly reported, and in general, is what practitioners care about in CFD analyses

---

> ### Author Response · Authors · 2025-12-03
> **New experiments with Erwin added**
>
> We thank the reviewer for carefully reading our manuscript. Our responses to the reviewer's questions/criticism are as follows:
>
> **The primary weakness is that the model does not outperform current baselines...**
>
> Note that we have now incorporated Erwin-Small/Medium with DO as a baseline, in the revised manuscript. We found Erwin’s performance to be very poor for the Ahmed dataset, and thus we decided to not include Erwin’s results on Ahmed. Since, we have only included Erwin’s results for Shapenet, thus we have decided to furnish Erwin’s results in the Appendix (Table 5). Although Erwin improves in terms of rmse, its coverage and calibration (as measured through MCAL, IS) is noticeably worse than PDN. When judged in terms of pure accuracy (if we ignore UQ for a moment), having considered both the datasets, PDN is still competitive with Transolver/Erwin on the Shapenet dataset, while it outperforms both models on Ahmed.  We like to however reiterate the reviewer’s comment that PDN is more than a mere architecture. DN’s inherent inductive biases for surface learning tasks, coupled with the proposed methodology for principled UQ, makes PDN a powerful model, particularly in small data regimes (as is often the case for many CFD applications). Maintaining highly accurate predictions with good quality UQ is a challenging task in which there is no match for PDN.  Finally, we like to also point out that a PDN block can be incorporated within an otherwise attention based architecture, like Transolver or PCT. This will potentially render such transformers — into a latent variable model enabling UQ.
>
> **..volumetric fields such as velocity/pressure are also important and are jointly predicted by other baselines..**
>
> (P)DN is restricted to closed surfaces, but it could still provide useful geometric inductive biases to predict quantities in the ambient domain, e.g. by attaching a GNO or cross-attention mechanisms to it. We will explore such extensions in future.
>
> **..I am curious about wall shear stress predictions...**
>
> P)DN can reconstruct and predict any number of surface-bound fields. However, we limit our experiments to only pressure fields, since none of the dataset we used includes reliable wall shear stress fields. The ShapeNet car dataset includes velocity fields, but the surface component is only available on a sparse set of points. We found the ground truth drag for the geometries in this dataset having a negligible effect of skin friction (obtained from the surface component of the velocity field), compared to the pressure drag. Nonetheless, we agree with reviewer that drag is commonly reported and thus we have reported the pressure drag for PDN in Table 8 (Appendix G). However, we opine that Lift is a very volatile quantity in CFD, and lot of careful consideration is required before endowing Lift prediction with UQ. We leave this as task for future explorations, and strongly believe that our work provides a platform for such explorations.
>
> We have also corrected the typos spotted by the reviewer in the revised manuscript.

---

### Official Review · Reviewer_nqZA · 2025-10-31

**Soundness:** 3
**Presentation:** 2
**Contribution:** 2
**Rating:** 4
**Confidence:** 4

**Summary:**

This paper introduces a new architecture, Probabilistic DiffusionNet (PDN), for learning uncertainty-aware surrogate models for Partial Differential Equations (PDEs) where the solutions depend on surface geometry. The work extends the existing DiffusionNet (DN) architecture which is a powerful model for learning surfaces, by reformulating the core deterministic diffusion mechanism of DN to be stochastic in nature. This stochastic diffusion operator is derived from the spectral solution of a stochastic heat equation (SPDE) and through this reformulation, the authors are compelled to inject spatially-correlated noise directly into the message-passing layers of their model. The architecture is constructed as a hierarchical probabilistic generative model similar to a Variational Auto-Encoder (VAE) where latent random variables $z$ are introduced at each layer. The model is fitted via amortized variational inference by maximizing the Evidence Lower Bound (ELBO).

**Strengths:**

The formulation of PDN as a latent variable model is sound. The authors clearly define the generative process $p_\theta(u|v, M^h)$, the observational model $p(y_s|u)$, and the amortized variational encoder $q_\phi(Z|Y,V,\mathcal{M})$. The use of the ELBO (Eq. 14) for optimization is a standard and appropriate choice for this framework. Central to the method—the stochastic diffusion operator $\mathcal{SP}_{t,\eta}$—there is nothing arbitrary. The operator is carefully derived from the spectral solution of the stochastic heat equation (Eq. (5)). This derivation is in Sec. 3.1 and proved in App. B which gives a rigorous mathematical justification for the construction of the model.

**Weaknesses:**

1. In Appendix C, the authors ablate two types of amortized encoders: "Partial Diffusion (PD)" and "DiffusionNet+ (DN+)". They state that the DN+ encoder is immune to the "basis ambiguity problem", whereas the PD encoder (used in the main paper) suffers from it. The authors choose the PD encoder because it has ~4x fewer parameters and achieves similar performance. This is a slightly unsatisfying trade-off. It is surprising that the theoretically less-robust encoder performs just as well. The paper would be stronger if it either used the more principled DN+ encoder or provided a deeper analysis of why the basis ambiguity issue, which they acknowledge, does not seem to degrade performance in this specific VAE framework.

2. The stochastic operator $\mathcal{SP}_{t,\eta}$ introduces a new and apparently important set of parameters, $\eta$, which shape the noise spectrum through the relation $q_k = \lambda_k^{2\eta}$. These parameters determine how strong and what kind of stochastic perturbations occur at each layer. However, the paper never clearly explains how the $\eta$ parameters are actually chosen or managed. Are these hyperparameters fixed or learned parameters that are optimized during training? If they are learned, an analysis of their converged values would be valuable. If they are fixed, an ablation study on their value is needed in order to determine the model’s sensitivity to this key UQ governing parameter.

3. The authors correctly identify the reliance on the eigendecomposition of the cotan-Laplacian as a limitation for scalability. This cost is paid offline, so it doesn't affect inference or training time (as reported in Table 7), but it is a significant pre-computation step that could become prohibitive for meshes with millions of vertices. While this is noted as future work, it is a practical barrier to the "industrial scale" applications the paper targets.

**Questions:**

1. The experiments are thoroughly conducted on two standard CFD benchmarks (ShapeNet car and Ahmed bodies), both involving the Reynolds-Averaged Navier-Stokes equations. While the results are strong, this focuses the paper's validation on a single class of physical problems (fluid dynamics). Could the authors comment on the expected generality of Probabilistic DiffusionNet? How readily would the proposed stochastic diffusion operator and its theoretical underpinnings (Theorem 3.1) apply to other types of PDEs on surfaces, such as problems in structural mechanics (e.g., shell elasticity) or electromagnetics (e.g., surface currents), which also rely heavily on geometric properties?

2. Could the authors please expand on the choice of the "Partial Diffusion" encoder? Given that the "DN+" encoder is more theoretically robust by avoiding basis ambiguities, what is the intuition for why it did not outperform the simpler PD encoder (Table 4)? Does the basis ambiguity in the PD encoder perhaps act as a form of regularization for the variational approximation, or is the issue simply not a practical concern for these datasets?

3. How are the noise spectrum parameters $\eta_l$ for each stochastic block $l$ (from Eq. 7 and 8) treated? Are they learnable parameters optimized jointly with $\theta$ and $\phi$? If so, do they converge to different values at different layers, and how does this affect the multi-scale uncertainty injection? If they are fixed hyperparameters, how were they selected, and how sensitive is the model's UQ performance (e.g., NLL or MCAL) to this choice?

4. The stochastic heat equation in Eq. (5), $\partial_t v = \Delta_M v + \mathcal{W}^Q$, appears to use a time-independent spatial noise field $\mathcal{W}^Q$. The mild solution in Eq. (1639) also suggests this. This formulation is slightly different from more common SPDE formulations that involve, for example, space-time white noise. Could the authors give the physical or mathematical justification for having chosen this particular form of SPDE? Was a formulation involving time varying noise considered? How might this affect the final stochastic operator $\mathcal{SP}_{t,\eta}$?

---

> ### Author Response · Authors · 2025-12-03
> **Revised manuscript updated with analysis of basis ambiguity, converged $\eta$**
>
> We thank the reviewer for carefully reading the paper and raising very thought questions/criticism. Our response to these are as follows:
>
> **Basis ambiguity of PD encoder**
>
> We like to point out first that there were some typos, leading to incorrect description of the PD encoder, which importantly missed a softplus parameterisation. We have now corrected this description in the revised manuscript. Furthermore, we have provided a comprehensive analysis of the effects of PD’s basis ambiguity in Appendix C.
>
> The primary effect of the basis ambiguity in the PD encoder is arbitrary sign flips of the eigenvectors across different meshes, for similar shapes. Despite the sign flips the mean of the variational distribution, and consequently the noise perturbation term (in Eq 7), by construction, is preserved. The variance nonetheless changes. However, due to the softplus parameterisation this variation (of the variance) remains bounded and continuous, preventing the extreme fluctuations that would destabilise training. The KL divergence term in the ELBO further regularises this behaviour by penalising large deviations from unit variance. This mechanism explains the empirical success of PD encoders, for these small datasets, as the basis ambiguity effectively acts as a form of stochastic regularisation during training (as correctly suspected by the reviewer), with the model learning to operate in regimes where sign variations have minimal impact on the overall reconstruction quality.
>
> **Analysis of converged noise spectrum parameters $\eta$**
>
> We are learning this parameter, one each per channel, per block. This is also clearly mentioned in the paper (see sentence above Eq 8, where we define $\theta$). In Appendix G.5 we have now provided plots of the histogram (across channels) of $\eta$, per block. The learned negative values of $\eta$, on average, across all channels, and per block, reveal that PDN has discovered a physically-motivated uncertainty structure. In the stochastic diffusion operator, the noise term $[\mathcal{N}{t,\eta,z}]{kc} = \lambda_k^{\eta_c-1}(1-e^{-\lambda_k t_c}) z_{kc}$ is weighted by $\lambda_k^{\eta_c-1}$, which with negative $\eta_c$ creates inverse spectral weighting: low-frequency modes with small eigenvalues receive large noise amplitudes, while high-frequency modes with large eigenvalues are strongly suppressed. This learned pattern indicates the model expresses uncertainty primarily about large-scale, smooth features rather than fine details.  Effectively, the stochastic diffusion operator (Eq 7) has learned to incorporate a low-pass stochastic perturbation, with similar scale across the layers.
>
> **Space-time SPDE formulation**
>
> The reviewer correctly noted that our formulation differs from the more common space-time white noise setup. We did indeed consider the time-dependent formulation with $\dot{W}^Q_t$, which leads to the SPDE: $\partial_tv(t) = \Delta_Mv(t) + \dot{W}^Q_t$. The solution on each eigenmode would be an Ornstein-Uhlenbeck process that, using the reparameterization trick, can be expressed as:
> $\hat{v}_k(t) = e^{-\lambda_k t}\langle v_0, \phi_k \rangle_{L^2(M)} + \sqrt{\frac{q_k(1-e^{-2\lambda_k t})}{2\lambda_k}} z_k$,
>  where $z_k \sim \mathcal{N}(0,1)$. Both formulations are mathematically valid and computationally equivalent, reducing to sampling from Gaussians with different variance structures. The time-independent version simplifies the presentation without sacrificing modelling capability.
>
> **Eigendecomposition of the cotan-Laplacian as a limitation for scalability**
>
> We have updated Appendix G.3 to include information about this aspect and specific timing information. Note that the cotan-Laplacian of a closed 2-manifold is a PSD sparse matrix, and so is amenable to preconditioning methods to accelerate eigen-decomposition. We employ a Cholesky preconditioning allowing us to eigen-decompose Laplacians for meshes of ~10^6 nodes in <6s on GPU, rendering the method suitable for typical industrial applications.
>
> **paper's validation on a single class of physical problems (fluid dynamics).**
>
> P(DN) is applicable to other phenomena of interest on surfaces. Firstly, note that the problem we introduce in §2.1 does not make any assumptions about the particular PDE that governs the ambient field, and so solutions on the boundary for other PDEs can be targeted with this approach (e.g. Maxwell’s equations, Burgers’ equation, etc.). However, prediction accuracy is expected to decrease for systems that are more chaotic or highly sensitive to changes in boundary conditions (especially on closed 2-manifold geometries). Secondly, while our current problem formulation excludes PDEs on the manifold (e.g., shell elasticity), empirical evidence and partial theory suggest the DN architecture is also suitable for such problems. We reserve the full technical argument, including universal approximation properties, for a future venue.

---

### Official Review · Reviewer_BNtL · 2025-11-03

**Soundness:** 3
**Presentation:** 3
**Contribution:** 2
**Rating:** 2
**Confidence:** 4

**Summary:**

The paper proposed a probabilistic version of DiffusionNet learning the map from the irregular geometries and boundary conditions to the PDE solution. The key idea is to replace the deterministic diffusion operator in DiffusionNet by a stochastic diffusion operator, which is equivalent to placing a prior over the message passing step. Then the paper uses variational inference to estimate the predictive distribution of the solution field.

**Strengths:**

1. The paper is written clearly.
2. The method looks sound.

**Weaknesses:**

1. The motivation is a bit skeptical. There can be many simpler choices for learning a probabilistic version of DiffusionNet to enable UQ. The paper proposes a rather complex framework --- injecting noises via SDE in message passing, and performing amortized VI --- which does not seem a strong necessity. For instance, why not use MC dropout, Deep Ensemble, Laplace method, or SWA-Gaussian (SWAG)? These methods all require minor or even no modification of the existing model/training, with a little bit extra work. From the methodology perspective, it is unclear where the advantage of the proposed method stands out, in addition to extra complexity.

2. Empirical performance is not supportive. The experimental results do not demonstrate the proposed method can largely improve simpler alternatives. Looking at Table 2, the performance of DN-DO/LA/ME is often comparable to or even better than PDN. It strengthens the doubt about the meaning/necessity of PDN --- why not we retreat to these classical, simple yet also powerful probabilistic training method? Why should we develop a new method?

3. The experiments are not sufficient. Only two datasets are employed for testing, which is below the bar in this community. In addition, there is no standard deviations and significance analysis, making it hard to conclude whether or not  there is a difference between PDN and competing methods.

**Questions:**

see above

---

> ### Author Response · Authors · 2025-12-03
> **Experiments clearly show that PDN is the only method that can produce high quality UQ without sacrificing accuracy**
>
> We thanks the reviewer for going through our work. However, we strongly, but respectfully, disagree with the reviewer's overall criticism of our work. We summarise our responses in the following:
>
> **The motivation is a bit skeptical..**
>
> For any UQ method to be useful in a practical and robust way, for industrial applications, must satisfy the following three conditions:
> i) The accuracy should not drop with respect to the deterministic performance of the same model, ii) the quality of UQ should be good, as measured through multiple metrics for probabilistic predictions and iii) should be computationally scalable. There isn’t any UQ method that ensures all three (when used with a base model that is generally used for resolution invariant PDE surrogacy like neural operators), including the ones the review mentions. It is generally found (see for example [1]) that ME can satisfy i) and ii) but obviously fails in iii). We design PDN to be a one stop solution for all three, and our results do reinforce that claim.
>
> Regarding the suggestion that other methods require minor or even no modification of the existing model/training” or that PDN introduces “extra complexity”, we respectfully disagree with these statements. Laplace/SWAG/ME needs additional compute so the statement about complexity is simply not true. Although none of these UQ methods change the base model, some of these require significant post-training (Laplace/SWAG) tuning to extract UQ, including the fact that the results are very sensitive to hyper parameters. As an example, Laplace requires approximating the hessian of a loss function with respect to model parameters post training. With models of over a million parameters, such an approach is extremely computationally expensive. In contrast, PDN follows a VAE training/inference framework (a widely and standard approach across machine learning) with little additional compute than DN.
>
> **Empirical performance is not supportive**
>
> We like to point out, respectfully, that this is an incorrect reading of the results. A correct judgement of the performance of a UQ method requires a holistic assessment of the quality of probabilistic predictions, ideally through multiple metrics each of which quantifies a particular aspect of the predictive distribution. The ones we provide (IS, MCAL, and NLL) are quantifying these different aspects, thus a method needs to be good in terms of all these metrics, and across different datasets. Conversely, if a method performs particularly poorly in one or more of the metrics then that reveals a pathology. Additionally, a significant concern for adopting off-the-shelf UQ methods is the drop in accuracy. A practitioner would not want to compromise on accuracy while switching from deterministic to probabilistic predictions. We can clearly see from Table 2 that DN-DO performs poorly across the board, worst performing method out of all (including GINO/TS). DN-LA produces significantly worse NLL, IS (which reveals a pathology, significant overestimation of uncertainty) for ShapeNet cars. Finally, DN-ME although produces similar or better NLL/IS/MCAL, is significantly worse in terms of accuracy, again revealing a pathology. PDN on the other hand produces a balanced performance, which is consistent across datasets. Furthermore, note that only DO is similar in terms of computation to PDN, since LA needs additional computation for Hessian and ME is 10x PDN's compute.
> When evaluated holistically, as we argue here, then PDN is clearly superior to the other methods (DN + DO/LA/ME), as shown in Table 2.
>
> **The experiments are not sufficient. **
>
> Our fundamental goal is to build a surrogate model for industrial CFD applications that takes as input varying geometry and predicts a surface field, corresponding to each geometry. In recent literature such models have been tested with two datasets. We rely on and cite [2,3,4] to motivate the choice of experiments — these are influential papers used as baselines, which use exactly the two datasets that we have used.
>
> ** In addition, there is no standard deviations .. **
>
> In Table 2 we show the average (across three repeats) of each metric, for both datasets, except ones obtained by ME. We have now included the metrics for each repeats in the Appendix (Table 5 -8) in the revised manuscript.
>
> [1] Li et al. Multi-resolution active learning of Fourier neural operators. AISTATS, 2024.
>
> [2]  Li et al. Geometry-informed neural operator for large-scale 3d PDEs. Neurips, 2023.
>
> [3] Pepe et al. Fengbo: a Clifford neural operator pipeline for 3d PDEs in computational fluid dynamics. IICLR, 2025
>
> [4] Deng et al. Geometry-guided conditional adaptation for surrogate models of large-scale 3D PDEs on arbitrary geometries. IJCAI, 2024

---

### Meta-Review · Area_Chair_rxe8 · 2026-01-06

**Summary:**

The reviewers raised the following concerns:

1. Reviewer BNtL raises concern that there are many simpler choices for learning a probabilistic version of DiffusionNet to enable UQ.

2. Reviewer BNtL raises concern that experimental results do not demonstrate the proposed method can largely improve simpler alternatives. Reviewer ioK4 commented that the proposed method does not outperform current baselines.

3. Reviewers nqZA and BNtL commented that only two datasets are used and not sufficient.

4. Several specific questions raised by Reviewer nqZA

**Reviewer Concerns:**

After the rebuttal, I think the above concerns 3 and 4 are resolved. For concerns 2, although the authors have made a good point that the proposed method provides a more well-rounded results, from the Table 2 I can still see some baselines such as TS-ME that offers similar overall performance. Therefore, the above concern 2 is not fully addressed. Furthermore, the paper can benefit from a clearer presentation and more thorough experiments.

**Reviewer Scores:**

After the rebuttal, I think reviewer BNtL, ioK4, and 1PJd will remain their score of 2, 6, and 8 respectively. Reviewer nqZA may raise the score to 6.

---

### Decision · Program_Chairs · 2026-01-26

Reject